# CHAIN-OF-ACTION: FAITHFUL AND MULTIMODAL QUESTION ANSWERING THROUGH LARGE LANGUAGE MODELS

**Zhenyu Pan**[†]  **Haozheng Luo**[†]  **Manling Li**[†]  **Han Liu**[†♮]
[†] Department of Computer Science, Northwestern University, Evanston, IL 60208, USA
[♮] Department of Statistics and Data Science, Northwestern University, Evanston, IL 60208, USA
{zhenyupan, hluo}@u.northwestern.edu   {manling.li, hanliu}@northwestern.edu

## ABSTRACT

We present a Chain-of-Action (CoA) framework for multimodal and retrieval-augmented Question-Answering (QA). Compared to the literature, CoA overcomes two major challenges of current QA applications: (i) unfaithful hallucination that is inconsistent with real-time or domain facts and (ii) weak reasoning performance over compositional information. Our key contribution is a novel reasoning-retrieval mechanism that decomposes a complex question into a reasoning chain via systematic prompting and pre-designed actions. Methodologically, we propose three types of domain-adaptable 'Plug-and-Play' actions for retrieving real-time information from heterogeneous sources. We also propose a multi-reference faith score to verify conflicts in the answers. In addition, our system demonstrates that detecting the knowledge boundaries of LLMs can significantly reduce both system latency and LLM usage in QA tasks. Empirically, we exploit both public benchmarks and a Web3 case study to demonstrate the capability of CoA over other methods.

## 1 INTRODUCTION

This work proposes a new reasoning-retrieval framework to enhance the quality of Large Language Models (LLMs) question answering without additional training and querying costs. As exemplified in Figure 1, this work overcomes three major drawbacks in applying LLMs to answer complex questions: (i) **unfaithful generation**, where the response may not align with real-time or domain-specific facts (e.g. failing to localize relevant facts in Figure 1(b)), (ii) **weak reasoning**, where LLMs struggle to aggregate heterogeneous information sources, resolve their conflicts, adequately reason over the information to provide useful, tailored responses (such as the failure of the stopped analysis in Figure 1(c) despite having successfully localized relevant search results), and (iii) **inefficient process**, where a large number of interactions with LLMs and token usage are costly.

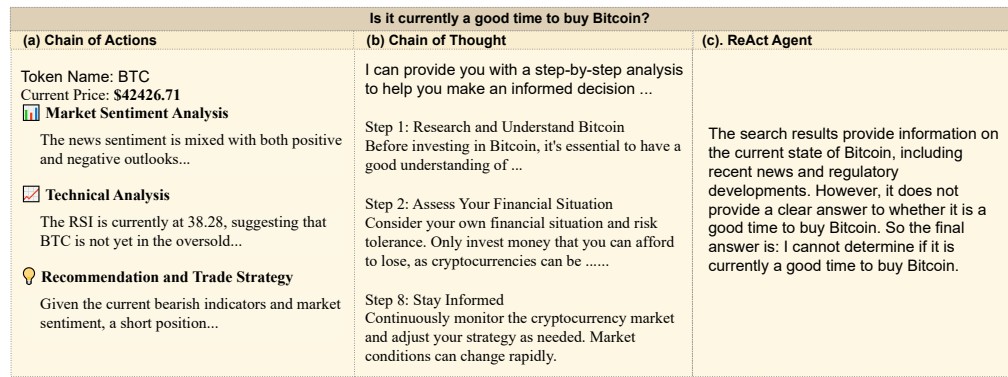

Figure 1: Chain-of-action prompting empowers LLMs to generate (1) faithful, informative, concrete analysis grounded in heterogeneous sources (open web, domain knowledge, tabular data, etc.) as well as (2) well-reasoned chains for complex questions to better interpret human goals and intents. This stands superior to previous approaches that yield generic, ambiguous, high-level responses.

Figure 2: Overview of Chain-of-Action framework. We use in-context learning to prompt LLM to generate the action chain. The chain has many nodes consisting of sub-questions (Sub), missing flags (MF), and LLM-generated guess answers (A). Then, the actions address multimodal retrieval of the nodes in three steps: (i) retrieving related information, (ii) verifying whether the LLM-generated answer needs correction by retrieval, and (iii) checking if we need to fill in missing contents with the retrieval. Finally, we generate the final answer by the LLM based on the processed action chain.

To enhance faithfulness and multi-step reasoning, previous approaches such as chain-of-thought based work [28, 21, 32, 30] encourage LLMs to think step-by-step to break down complex questions. However, only pushing models to continue thinking may not be ideal. Models are expected to learn to pause to verify results and decide if they need more information before continuing to generate. Recent work, thereby, explores integrating information retrieval [33, 31, 12] into the reasoning chain. However, we argue that seeking external information is not only retrieval, but should manifest as configurable **'Plug-and-Play' actions**: querying web text, encoding domain knowledge, analyzing tabular and numerical data, etc. The term 'plug-and-play' refers to the ability to freely add or remove pre-designed actions, such as the three different actions implemented in our work. However, for any new action to be integrated in the future, careful design and adjustment will be required to ensure compatibility with the framework's input and output formats. The key challenge of such heterogeneous data is to automatically decide when to cease generation to solicit information, what types of external sources to leverage, and how to cross-validate conflicting insights. In addition, previous efforts to improve system efficiency have primarily focused on accelerating the retrieval process. However, we observe that most of the cost lies in summarizing the retrieved information. By detecting the knowledge boundary—referring to the parametric knowledge that the model has acquired during its training on a high-quality dataset—we can reduce the need for frequent summarization. Since LLMs with extensive parameters have undergone costly training, leveraging this comprehensive knowledge can minimize unnecessary efforts during the filtering and summarization phases.

To that end, we propose a universal framework CoA equipping LLMs to proactively initiate information-seeking actions. We design three 'Plug-and-Play' actions in this paper: (i) **web-querying** to extract real-time information as discrete text tokens, (ii) **knowledge-encoding** to embed domain-specific knowledge concepts as continuous vectors, and (iii) **data-analyzing** for accessing and interpreting numeric tabular sources. A key advantage of this framework is the extensibility to diverse modalities, e.g., images in the future. Beyond adapting across data modalities, new actions can be introduced to handle emerging domains or data processing techniques. Additionally, our direct prompting strategy for detecting knowledge boundaries reduces both system latency and LLM usage.

In detail, as illustrated in Figure 2, the CoA first inject the question and action descriptions into the pre-designed prompting template through in-context learning. Then LLMs construct an **action chains (ACs)**, where each *action node* represents a sub-question, a missing-data flag indicating the need for additional information, and an initial answer. After that, we perform **action execution and monitoring** to address retrieval demands in three steps: (i) retrieving related information, (ii) verifying conflict between the initial answer and the retrieved information, and (iii) inferring missing content with retrieved information when necessary. To verify information conflicts, we design a verification module utilizing our **multi-reference faith score** (MRFS). If the generated answer confidence is below a threshold, the corresponding action incorporates the retrieved information for answer correction. In this way, LLMs can generate the final answer that is sound and externally-grounded. A key feature of CoA is automatically solicit external information that forms as tokens, vectors, or numbers for integration into model reasoning. Rather than hard-coding their connections, actions are designed as dataset-agnostic modules that LLMs invoke selectively.

The significant improvement of CoA is not only showed in experiments on multiple QA datasets, but also is validated from the success of the real-world deployment. Upon integration into a Web3 QA application, key metrics including active users and positive feedback volumes increased remarkably within a few months. This performance highlights CoA's effectiveness in real-world applications.

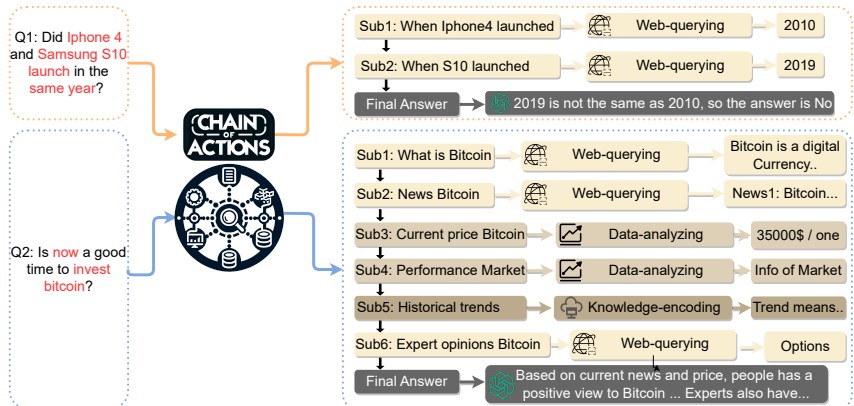

Figure 3: Two samples from our Chain-of-Action Framework.

In summary, our main contributions are as follows:

- We present CoA, which integrates a novel reasoning-retrieval mechanism to decompose complex questions into reasoning chains of configurable actions via systematic prompting. It can retrieve heterogeneous information and reduce information conflicts.
- We propose three types of 'Plug-and-Play' domain-adaptable actions to address retrievals for real-time information, domain knowledge, and tabular data. The actions are flexible to incorporate additional sources.
- We propose a novel metric, multi-reference faith score (MRFS), to identify and resolve conflicts between retrieved information and LLM-generated answers, enhancing reliability. It also significantly reduces both LLM interaction frequency and token usage.
- Our Web3 QA product shows significant user engagement and positive feedback, validating the CoA framework's effectiveness and practicality in real-world scenarios.

## 2 METHODOLOGY

In this work, we propose a 'Plug-and-Play' framework adaptable to different data modalities, currently capable of handling text and tabular data. The current focus is to validate the framework's efficacy in these two modalities, laying a solid foundation for further integration of additional modalities. This intention is also a significant direction for our future work involving Vision Language Models. As shown in Figure 2, we first introduce how to generate the action chain by LLM (Sec. 2.1). Then, the actions address multimodal retrieval demands of the chain's nodes in three processes: (i) retrieving related information, (ii) verifying whether the LLM-generated answer is good enough or in demand of more information from retrieval, and (iii) checking if the initial answer of each node's sub-question is missing so that we fill in missing contents with the retrieved information. Finally, we get the final answer by the LLM based on this refined and processed action chain.

### 2.1 ACTION CHAIN GENERATION

We use in-context learning to generate an action chain by LLM. As shown in Figure 2 (a), we design a prompt template to decompose the user's question into many sub-questions, as well as the corresponding *Missing Flags* (MF) and guess answers shown in Figure 2 (b). Then, we assign one of the actions to solve each sub-question.

**Prompt design.** We design a prompt template as shown in Figure 4 starting with "Construct an action reasoning chain for [questions]..." to prompt LLM to generate an Action Chain *AC* not answer our question *Q* directly.

$$
\begin{aligned}
AC_Q = &(\texttt{Action}_1, \texttt{Sub}_1, \texttt{MF}_1, \texttt{A}_1), \\
&\to (\texttt{Action}_2, \texttt{Sub}_2, \texttt{MF}_2, \texttt{A}_2), \ldots, \\
&\to (\texttt{Action}_n, \texttt{Sub}_n, \texttt{MF}_n, \texttt{A}_n).
\end{aligned}
\tag{1}
$$

Each action node represents four elements, including $\texttt{Action}_i$, the sub-questions $\texttt{Sub}_i$, the missing flag $\texttt{MF}_i$, the guess answer from LLMs $\texttt{A}_i$, where $i \in \{1, \ldots, n\}$. When the inner-knowledge of LLM is enough to answer the sub-questions, LLM generates an initial answer as the "guess_answer". Otherwise, the value of "missing_flag" becomes "True", followed by a blank "guess_answer".

**Construct an action reasoning chain** for this complex **[Question]**: "$QUESTION" in JSON format. For each step of the reasoning chain, **choose an action** from three choices: [Web-querying Engine(search real-time news or new words), Knowledge-encoding Engine (search existing domain information in local knowledge base), Data-analyzing Engine (query real-value data and calculate some results) ] as the value of element "Action", and also **generate a sub-question** for each action to get one of [web-search keywords, needed information description, data features description] as the value of element "Sub". Also, **generate an initial answer** for each Sub as the value of the element "Guess_answer" if you make sure it is correct. In addition, if you cannot answer some sub-questions, **make the element "Missing_flag" value** "False", otherwise, make it "True" You need to try to **generate the final answer** for the [Question] by referring to the **"Action"-"Sub"-"Guess_answer"-"Missing_flag" in "Chain"**, as the value of the element "Final_answer". For example:

{"Question": "Is it good to invest in Bitcoin now? A. It is a good time. B. It is not a good time.",
  "Chain": [
    {**"Action": "Knowledge-encoding", "Sub": "what is bitcoin", "Guess_answer": "Bitcoin is one of the cryptocurrencies.", "Missing_flag" : "False"**},
    {**"Action": "Data-analyzing", "Sub": "What is the recent price trend of bitcoin?", "Guess_answer": "The price of Bitcoin increases from XX to YY…", "Missing_flag" : "False"**},
    {**"Action": "Web-querying", "Sub": "bitcoin news", "Guess_answer": "", "Missing_flag": "True"**}],
  "Final_answer": "Bitcoin is one of the cryptocurrencies that is risky to invest [1]. And its price become more and more high recently [2]. Also, there is a lot of news to promote Bitcoin. So, it is a good time to invest in Bitcoin now."}

Figure 4: Prompt to Generate Action Chain in Chain-of-Action (CoA). This template integrates the user's question along with a description of each available action. The resulting action chain comprises elements such as actions, subs, guess answers and missing flags. This prompt not only decomposes complex questions into multiple sub-questions, guided by the features of the actions but also allows the LLM to answer certain sub-questions using its existing inner-knowledge. This process exemplifies our proposed reasoning-retrieval mechanism.

## 2.2 Actions Implementation

We propose three types of actions to address multimodal retrieval demands (text and tabular data): (1) Web-querying for searching real-time text data from websites, (2) Knowledge-encoding for retrieving related text data from local domain-specific corpus datasets, and (3) Data-analyzing for extracting tabular data from domain-specific databases via generated SQL codes. Each action has three steps to execute: (i) Information Retrieval, (ii) Answering Verification, and (iii) Missing Detection. We first introduce the design of the actions. Then, we describe the details of three common steps. When combined with textual data, we want to demonstrate that tabular data forms a critical part of heterogeneous inputs, significantly enhancing system understanding and response capabilities.

### 2.2.1 Data Collection

**Action 1: Web-querying.** Web-querying action utilizes the existing search engines (e.g., Google Search) and follows our query strategy to get the relevant content from the Internet. In detail, it first searches for the keywords of the given sub-question $\mathsf{Sub}_n$ to obtain the result list. If the corresponding "Missing_flag" is "True", we choose the top-k results and extract their contents from their page sources. Otherwise, we combine their titles $\mathsf{T}$ and snippets $\mathsf{Sn}$ of the top M pages. Then, we transfer each pair of title and snippet $\{\mathsf{T}_m, \mathsf{Sn}_m\}$ into a 1536-dimension vector $Emb\{\mathsf{T}_m|\mathsf{Sn}_m\}$ by the embedding model (text-embedding-ada-002 from OpenAI [17]). Meanwhile, we also transfer the sub-question and guess answer $\{\mathsf{Sub}_n, \mathsf{A}_n\}$ into $Emb\{\mathsf{Sub}_n|\mathsf{A}_n\}$. Next, we calculate the similarity between each $Emb\{\mathsf{T}_m|\mathsf{Sn}_m\}$ and $Emb\{\mathsf{Sub}_n|\mathsf{A}_n\}$ to filter the pages whose similarities are lower than 0.8. Then, we extract the contents of high-similarity pages and calculate the similarity between them and $Emb\{\mathsf{Sub}_n|\mathsf{A}_n\}$ to rank and get the top-k final pages. Those contents of the k final pages are the final information that we retrieve by the action.

**Action 2: Knowledge-encoding.** Knowledge-encoding action utilizes the vector database (e.g., ChromaDB) as data storage to store the domain information and corresponding embedded vectors. For example, we collect web3 domain information from different sources (X, experts' blogs, white papers, and trending strategies) to support our QA case study. After data collection, we split each document into many chunks based on the length. Then, we encode each chunk of content into an embedded vector and store it in our vector database with its index. When we need to execute this engine to retrieve domain information, we could forward the $Emb\{\mathsf{Sub}_n|\mathsf{A}_n\}$ to compute the similarity between the input and each chunk to obtain the top-k results.

**Action 3: Data-analyzing.** Data-analyzing action aims to retrieve the data information from some real-value data sources (e.g., market data of digital currencies). In some special situations, we could directly retrieve the relevant values from our deployed API when some sub-questions demand up-to-date or historical value data. Furthermore, we can also use LLM to compute more sophisticated features by generating Python or SQL codes to execute. It is flexible and compatible with various situations. In this paper, we only design it to retrieve the market data for the Web3 case.

### 2.2.2 Data Verification

In the action chain, the framework executes the data collection for each node until it finishes the whole chain, as shown in Algorithm 1.

**Information Retrieval.** In the information retrieval stage, we need to find the most relevant and similar contents from different knowledge/data sources. At first, we choose both sub-questions and guess the answer of each node as a query section, $QS_n$. Then, with the encoding of LLM's embedding model, we transfer our query $QS_n = \{\mathsf{Sub}_n | A_n\}$ into a 1536-dimension vector $Emb\{QS_n\}$. With this embedded vector, we can perform information retrieval and then rank the results by calculating the similarity. Finally, actions return the top-k results $R_{\{QS\}}$:

$$R_{\{QS\}} = (r_1 \,|\, r_2 \,|\, ... \,|\, r_k). \tag{2}$$

**Answering Verification.** After the information retrieval, we verify the information conflicts between guess answer $A_n$ and retrieved facts $R_{\{QS\}}$. Inspired by the ROUGE [13], we propose the MRFS. To get the MRFS, we compute the pairwise faith score $S$ between a candidate summary and every reference, then take the maximum of faith scores. $S$ is a composite metric computed based on three individual components: Precision (P), Recall (Rcl), and Average Word Length (AWL) in the Candidate Summary. The mathematical representation of the score is given by:

$$\mathrm{S} = \alpha \times P + \beta \times Rcl + \gamma \times AWL \tag{3}$$

Where:

- $\pmb{\alpha, \beta, \gamma}$ are weights corresponding to the importance of Precision, Recall, and Average Word Length, respectively. Their values can be adjusted based on specific requirements but should sum up to 1 for normalization purposes.
- $\pmb{P}$ (Precision) is the ratio of relevant tokens among the retrieved tokens:

$$P = \frac{\text{number of relevant items retrieved}}{\text{total number of items retrieved}} \tag{4}$$

- $\pmb{Rcl}$ (Recall) is defined as the ratio of relevant tokens that were retrieved:

$$Rcl = \frac{\text{number of relevant items retrieved}}{\text{total number of relevant items}} \tag{5}$$

- $\pmb{AWL}$ (Average Word Length in Candidate Summary) represents the mean length of the words present in the summarized content:

$$AWL = \frac{\text{sum of lengths of all words}}{\text{total number of words}} \tag{6}$$

Adjusting the weights $\alpha, \beta, \gamma$ will allow for emphasizing different aspects (Precision, Recall, or Word Length) depending on the specific evaluation criteria or context. After getting the MRFS through: $MRFS = \arg_k \max S(r_k, A_i)$, we setup a threshold $T$ to decide whether the answer $A_i$ is faithful. If MRFS is greater than T, we keep the answer; otherwise, we change the answer $A_i$ to reference contents. As shown in Fig 5, given a generated text $A_i$ ("david had an apple and a banana") and a reference text $r_k$ ("david is a good person, and he got an apple, a banana, and oranges."), we first tokenize both texts: $A_i$ tokens: ["david", "had", "an", "apple", "and", "a", "banana"] and $r_k$ tokens: ["david", "is", "a", "good", "person", "and", "he", "got", "an", "apple", "a", "banana", "and", "oranges"]. Then, for precision (**P**), we calculate the ratio of tokens in $A_i$ that are also found in $r_k$, the intersection of tokens are ["david", "an", "apple", "and", "a", "banana"] and the result is $P = \frac{6}{7} \approx 0.857$. For recall (**Rcl**), we calculate the ratio of tokens in $r_i$ that are correctly predicted by $A_i$ and get $Rcl = \frac{6}{14} \approx 0.429$. For average word length (AWL), we calculate by $AWL = \frac{5+3+2+5+3+1+6}{7} \approx 3.57$.

**Missing Detection.** The last stage of each action is detecting whether the guess answer $A_i$ is complete. When a sub-question needs some special or real-time information, the corresponding guess answer $A_i$ could be incomplete with a Missing Flag $MF_i$ being "true". If a guess answer's MF is "True", we inject the retrieved information into the $A_i$ to fill in the blank "Guess_answer".

### 2.3 FINAL ANSWER GENERATION

After all actions' executions, we use a prompt template shown in Figure 6 to integrate all corrected answers and sub-questions of the AC. Then, we prompt LLM with retrieved information and generate the final answer starting with "[Final Content]" through the corrected reasoning chain.

```
# Given texts:
# A (generated text) = "david had an apple and a banana"
# R (reference text) = "david is a good person, and he got an apple, a banana, and oranges."

# We need to calculate Precision (P), Recall (Rcl), and Average Word Length (AWL) using tokens as units.

# Break down the sentences into tokens
A_tokens = "david had an apple and a banana".split()
R_tokens = "david is a good person, and he got an apple, a banana, and oranges.".split()

# Calculate Precision and Recall
# Precision: Proportion of relevant instances among the retrieved instances
# Recall: Fraction of relevant instances that were retrieved

# Relevant tokens are those that appear in both A and R
relevant_tokens = set(A_tokens).intersection(R_tokens)

# Precision Calculation
precision = len(relevant_tokens) / len(A_tokens)

# Recall Calculation
recall = len(relevant_tokens) / len(R_tokens)

# Average Word Length (AWL) in Candidate Summary
# AWL is calculated as the sum of the lengths of all words divided by the total number of words in A
awl = sum(len(word) for word in A_tokens) / len(A_tokens)

precision, recall, awl
```

Figure 5: The pseudo codes about how to calculate the MRFS.

Table 1: The functional comparison of Chain-of-Thought baselines with our method CoA.

| Method | Few -shot | CoT | SC | ToT | Auto -CoT | Least- to-Most | ToolFormer | Self-Ask | React | DSP | SearchChain | CoA |
|---|---|---|---|---|---|---|---|---|---|---|---|---|
| Multistep Reasoning | ✓ | ✓ | ✓ | ✓ | ✓ | | | ✓ | ✓ | ✓ | ✓ | ✓ |
| Retrieval | | | | | | | ✓ | ✓ | ✓ | ✓ | ✓ | ✓ |
| Multimodal | | | | | | | | | | ✓ | | ✓ |
| Verification | | | | | | | | | ✓ | | ✓ | ✓ |

## 3 EXPERIMENTS

In this section, we compare the performance of our Chain-of-Action framework with state-of-the-art baselines across public benchmarks. Subsequently, we provide a detailed analysis of our launched case study: a Question Answering (QA) application in the Web3 domain.

### 3.1 EXPERIMENTS WITH BENCHMARKS

**Datasets** We select 4 classic, 1 long-form, and 1 open-domain QA task. Four **classic QA** tasks that include web-based QA (WQA) [2], general QA[1] (DATE, General Knowledge, Social QA (SoQA)), Truth QA [24], Strategy QA (SQA) [6], and Fact Checking (FEVER [26]). **Long-form QA** task is the first long-form QA dataset focusing on ambiguous factoid questions, ASQA [25]. **Open-domain QA** task is QReCC [1], testing the ability to handle context-dependent queries across different domains.

**Metrics** Cover-EM [20] are often used to represent whether the generated answer contains the ground truth. However, it has many limitations inherent in token-based exact-match scoring methods, which may not capture a model's understanding of complex queries and can lead to misjudgements. So, we propose GPT-EM to overcome the limitations in our evaluation process, aiming to provide a more accurate and nuanced assessment of performance across different prompting and agent framework. The prompt of GPT-4 that we use is shown in Appendix D. Besides GPT-EM, we also use ROUGE-L in ASQA. Unlike the ROUGE metric in Table 2, which serves as a baseline for our MRFS verification metric, ROUGE-L in ASQA focuses on the Longest Common Subsequence (LCS) to assess fluency and coherence in diverse long-form answers.

**Baselines** We have two types of baselines: the first type focuses on reasoning, prompting LLM to solve complex questions (Few-shot Prompting, Chain-of-Thought (CoT) [29], Self Consistency (SC) [28], Tree of Thought (ToT) [32], Least-to-Most [38], and Auto-Chain-of-Thought (Auto-CoT) [37]), and the second Retrieval-Augmented-Generation (RAG) type that integrates Information Retrieval to enhance reasoning capabilities (ToolFormer [22],Self-Ask [18], React [34], SearchChain (SeChain) [31], and DSP [9]). We conduct a thorough functional comparison between these baseline methods

---
[1] https://github.com/google/BIG-bench

Table 2: We conduct an evaluation of accuracy for 7 classical QA, 1 fact-checking dataset, and 1 long-form QA. Our study involves the 11 baseline methods alongside our CoA method. We assess the performance of these methods across seven tasks, considering both information retrieval and non-retrieval scenarios. The results average over three runs, are presented with variance values omitted (all $\leq 2\%$). Our presentation format involves bolding the best results and underlining the second-best results. Our findings highlight the superior performance of CoA, which achieved the highest accuracy in 12 out of 14 test scenarios. Notably, CoA consistently outperforms all baseline methods, even when external memory was not employed, demonstrating its robust and top-tier performance. Black means GPT-EM, and Red means ROUGE-L

| | Question Answering | | | | | | | Fact Checking | Long Form |
|---|---|---|---|---|---|---|---|---|---|
| Method | Web | DATE | GK | Social | Truth | Strategy | QReCC | FEVER | ASQA |
| Without Information Retrieval | | | | | | | | | |
| Zero-shot | 43.0 | 43.6 | 91.0 | 73.8 | 65.9 | 66.3 | 18.4 | 50.0 | 30.2/17.4 |
| Few-shot | 44.7 | 49.5 | 91.1 | 74.2 | **68.9** | 65.9 | 18.4 | 50.7 | 34.5/20.9 |
| CoT [29] | 42.5 | 43.7 | 88.1 | 71.0 | 66.2 | 65.8 | 30.6 | 40.4 | 47.4/21.1 |
| SC [28] | 36.5 | 50.0 | 87.5 | 60.0 | 66.7 | 70.8 | 67.4 | 53.3 | 34.8/20.3 |
| ToT [32] | 32.3 | 47.1 | 85.1 | 68.5 | 66.6 | 43.3 | 20.4 | 41.2 | 32.5/10.4 |
| Auto-CoT [37] | 42.1 | 52.3 | 89.7 | 59.1 | 61.6 | 65.4 | 21.0 | 32.5 | 36.3/21.0 |
| Lest-to-Most [38] | 44.0 | 42.1 | 80.8 | 68.1 | 59.5 | 65.8 | 22.4 | 43.4 | 39.1/23.7 |
| SeChain w/o IR | 50.8 | 44.7 | 75.0 | 64.9 | 54.1 | 75.6 | 39.2 | 35.9 | 35.7/16.4 |
| CoA w/o actions | **64.7** | **55.3** | **91.4** | **80.2** | 63.3 | 70.6 | 37.2 | **54.2** | 47.9/22.6 |
| Interaction with Information Retrieval | | | | | | | | | |
| ToolFormer [22] | 34.5 | 53.9 | 72.3 | 48.1 | 57.5 | 69.4 | 50.0 | 60.2 | 37.1/20.5 |
| Self-Ask [18] | 31.1 | 55.1 | 79.7 | 52.1 | 60.5 | 67.7 | 34.5 | 64.2 | 32.5/15.3 |
| React [34] | 38.3 | / | 85.1 | 65.8 | 59.9 | 70.4 | 37.3 | 43.9 | 45.0/18.8 |
| DSP [9] | 59.4 | 48.8 | 85.1 | 68.2 | 58.4 | 72.4 | 38.1 | 62.2 | 55.0/13.1 |
| SearchChain [31] | 65.3 | 51.0 | 87.6 | 69.4 | 61.7 | 77.0 | 57.3 | 65.9 | 45.2/21.9 |
| CoA (MRFS in verification) | **70.7** | **57.4** | **98.6** | **83.1** | **67.3** | **79.2** | **69.7** | **68.9** | **60.9/29.1** |
| -w/o verification | 66.9 | 56.8 | 95.7 | 81.5 | 65.0 | 75.2 | 66.3 | 65.7 | 55.7/24.0 |
| -w/o imputation | 67.4 | 56.3 | 97.1 | 82.9 | 65.8 | 76.5 | 63.8 | 65.3 | 54.9/24.3 |
| -w/ ROUGE | 68.3 | 56.9 | 96.2 | 81.9 | 65.7 | 76.3 | 67.2 | 67.2 | 56.2/26.8 |
| -w/o Action 1 | 65.0 | 55.9 | 89.3 | 81.5 | 64.2 | 75.8 | 51.5 | 65.3 | 55.2/25.0 |
| -w/o Action 2 | 68.1 | 56.3 | 95.2 | 82.5 | 65.5 | 76.2 | 66.8 | 67.0 | 55.9/26.0 |

Here is the **corrected reasoning chain** for this complex [**Question**]: "{how's buying bitcoin}". Each step of the reasoning chain has **[Sub-question] and [Solved Answer]**. Answer the [Question]: "{how's buying bitcoin}" starting with [Final Content] **through the reasoning chain**.

For example:
[Question]:" Is it good to invest bitcoin now?"
**[Sub 1]: What is bitcoin?**
**[Solved Answer]: Bitcoin is one of the digital concurrencies…**
**[Sub 2]: What is the recent price trend of bitcoin?**
**[Solved Answer]: the price of Bitcoin increases by 3018 dollars in…**
**[Sub3 ]: news of bitcoin**
**[Solved Answer]: One news shows that … And it promotes Bitcoin a lot…**
[Final Content]: Bitcoin is one of the cryptocurrencies that is risky to invest [1]. And its price become more and more high recently [2]. Also, there is a lot of news to promote Bitcoin such as… [3]. So the answer is It is a good time to invest in Bitcoin now, but you need to consider the risk of investing in cryptocurrency.

Figure 6: Prompt for final answer generation. We use the processed chain to prompt LLM to reanswer.

and our Chain-of-Action (CoA), as presented in Table 1. The "-w/o imputation" means that we do not use retrieved information to fill the blank answers of sub-questions that LLM cannot answer.

**Implementation.**    Our experimental framework integrates data preprocessing inspired by Big-Bench [24] and Auto-COT [37]. In generating process, all baselines (including CoA) use gpt-3.5-turbo [16] as the backbone. To address challenges in controlling response formats when working with black-box models, we developed an advanced evaluation pipeline that leverages GPT-4 [3]. GPT-4 is used exclusively in the evaluation stage to assess the alignment of the generated answers from all baselines (including CoA) with the ground truth. Details of the evaluation prompt used can be found in Appendix D. This setting ensures fairly comparison among all baselines while utilizing GPT-4's advanced evaluation capabilities to ensure consistent and reliable assessment.

Table 3: Usage comparison across benchmarks. Each block labeled as A+B/C represents the number of tokens used for the input (A), the output (B), and the time consumed in seconds (C) by the LLM.

|            | SQA                | WQA               | SoQA               | FEVER             |
|------------|--------------------|-------------------|--------------------|-------------------|
| Self-ask   | 811+15/0.87        | 805+19/1.03       | 881+21/1.03        | 860+19/0.88       |
| ReAct      | 73651+2920/99.8    | 20273+1331/65.3   | 47951+954/156.6    | 37409+1858/67.7   |
| SeChain    | 82120+3215/132.9   | 45388+1530/96.4   | 61097+1190/239.1   | 45561+2072/105.3  |
| CoA (ours) | **30485+1120/43.1**| **11873+605/35.6**| **26011+429/58.3** | **18824+1021/30.5**|

### 3.1.1 EXPERIMENTAL ANALYSIS

**Overall Performance.** Table 2 compares the effectiveness of our CoA framework and 11 baseline methods across 7 classical QA, 1 fact-checking, and 1 long-form QA datasets. We evaluate the performance in both information retrieval and non-retrieval scenarios, separately. The sole exception pertains to React, implemented by Langchian [27]. It exhibits an unresponsive behavior in the DATE dataset. As a result, we omit the comparison involving React within the DATE dataset. Our CoA framework demonstrates superior performance metrics in 12 of 14 test scenarios. Our method achieves a significant 3.42% improvement in the test tasks without information retrieval compared to the state-of-the-art baseline (SearchChain without IR), and a 6.14% increase in the test tasks with information retrieval over its state-of-the-art baseline (SearchChain). This is a significant outcome, as it underscores the effectiveness of our framework. It also demonstrates that CoA is well-suited for various question-answering tasks. In particular, the enhancement in performance is consistent regardless of the integration of IR. This indicates that our framework has intrinsic robustness and comprehensive understanding that is not reliant on external information. We also find our CoA without IR is impressive. After deeply explore how the process of generating and answering sub-questions contributes to these improvements in Appendix E, we get:

*Comparative Insights: Our CoA offers a richer analysis by integrating multiple aspects of the scenario into a comprehensive reasoning chain. This approach addresses the direct question and contextualizes Alex's actions within his duties and the prison's operational protocols, providing a multidimensional understanding. In contrast, CoT tends to a more straightforward, surface-level interpretation. This explanation underscores how CoA's approach provides deeper, more contextually enriched answers compared to CoT, making it particularly effective for complex scenarios requiring a nuanced understanding of actions within their broader social and procedural context.*

**Complexity of reasoning processes.** In a further analysis in Table 7, we delve into the complexity of reasoning processes in baselines. Our framework exhibits a higher average number of reasoning steps when decomposing complex questions. This metric is vital, highlighting the framework's capability to engage in a multi-step inference process, a capability that is essential for solving intricate problems that require more than surface-level understanding. The fact that our framework outperforms others in this measure suggests that it can better understand and navigate the layers of complexity within questions, which is a testament to the sophisticated reasoning algorithms it employs.

**Efficiency of current framework.** Table 3 and Table 4 explores the average system latency and LLM usage per question. We choose these 4 datasets compared to 8 datasets in Table 2 because they represent more complex tasks compared to the others. These datasets require higher number of reasoning steps and exhibit a greater misleading ratio by IR, which presents a more challenging benchmark. We believe focusing on complex datasets allows us to more distinctly demonstrate the superior performance of CoA in scenarios that are not only more demanding but also more indicative of real-world applications. CoA shows a reduced cost, reflecting the CoA's efficiency in minimizing system latency and LLM usage. It is a vital attribute for addressing complex issues with lower expenditure. It suggests that CoA surpasses others with **detecting knowledge boundaries** of LLM.

**Misleading of external knowledge.** Table 5 scrutinizes the methods in terms of their susceptibility to being misled by external knowledge. This is a nuanced aspect of framework evaluation, as it speaks to the framework's ability to discern relevant from irrelevant information, a nontrivial task in the age of information overload. Our framework emerges as the most resistant to misinformation, maintaining high accuracy even when interfacing with external data sources. This reveals not only the advanced data parsing and filtering capabilities of CoA but also its potential to mitigate the risks associated with the proliferation of false LLM-generated information.

**Robustness of current framework.** The verification of the correctness and relevance of decomposed sub-queries and model decisions is important. We employ a rigorous evaluation Actions framework.

Table 4: We perform a thorough analysis to compare the average number of interactions with LLM across four datasets. The results, obtained through three separate runs, are displayed without including variance values (all $\leq 0.4\%$).

|         | WQA | SQA | SoQA | FEVER |
|---------|-----|-----|------|-------|
| Self-Ask | 5.3 | 5.0 | 5.0 | 5.1 |
| React   | 5.2 | 5.2 | 5.3 | 5.5 |
| SeChain | 6.4 | 6.7 | 6.0 | 5.6 |
| CoA     | 4.0 | 4.0 | 4.4 | 4.2 |

Table 5: We perform an analysis showing that external knowledge leads LLM astray in answering using baseline methods. Our study takes place in a context involving information retrieval tasks.

|         | WQA | SQA | SoQA | FEVER |
|---------|-----|-----|------|-------|
| Self-Ask | 14.3 | 10.3 | 14.1 | 10.7 |
| React   | 16.1 | 10.0 | 15.8 | 11.2 |
| DSP     | 13.5 | 9.2 | 14.3 | 10.1 |
| SeChain | 7.2 | 5.3 | 9.4 | 8.5 |
| CoA     | 1.9 | 2.6 | 6.1 | 3.4 |

Table 6: Comparison of SOTA baselines—React' (Rt) and Self-Ask' (SA)—with our CoA method in the Web3 case, evaluated on coverage, non-redundancy, and readability. The results, averaged over three runs, are displayed without including variance values (all $\leq 0.4\%$). For the CoA results, the left side is the performance without tabular action, while the right side is the full version.

|               | Rt | SA | DSP | CoA |
|---------------|-----|-----|-----|---------|
| Coverage      | 1.5 | 1.8 | 1.7 | 2.0/2.9 |
| Non-redundancy | 2.0 | 1.9 | 2.1 | 2.2/2.3 |
| Readability   | 2.1 | 2.1 | 2.0 | 2.5/2.7 |
| Overall       | 1.9 | 2.0 | 2.0 | 2.0/2.6 |

Table 7: We conduct an analysis of the average number of reasoning steps to demonstrate the intricacy of test tasks. Our study takes place in a non-information retrieval context. The results, obtained through three separate runs, are displayed without including variance values (all $\leq 0.1\%$).

|               | WQA | SQA | SoQA |
|---------------|-----|-----|------|
| CoT           | 2.2 | 2.1 | 2.4 |
| SC            | 2.1 | 2.1 | 2.8 |
| Auto-CoT      | 3.2 | 2.9 | 3.0 |
| Least-to-Most | 1.2 | 1.2 | 1.8 |
| Self-Ask w/o IR | 2.1 | 2.4 | 2.9 |
| SeChain w/o IR | 3.4 | 3.7 | 4.0 |
| CoA w/o Actions | 3.9 | 4.1 | 4.6 |

To ensure a robust assessment, we randomly selected 200 questions from the Social QA dataset, which represents a maximum average number of reasoning steps. We then generated action chains for each question and assessed the correctness and relevance of the resulting sub-questions. We employed human annotation to evaluate the results, which yielded a high ratio of correctness at 95.5% and relevance at 97.0%. These results indicate a strong alignment between the decomposed sub-queries, the model decisions, and the expected outcomes. Furthermore, recognizing the importance of scalability and efficiency in evaluations, we are exploring the potential for automating the process of assessing relevance and correctness. This includes a thorough literature review to identify and integrate more explicit methods into our experimental framework, enhancing our ability to verify sub-questions and actions.

In conclusion, the empirical evidence from our assessments presents a compelling case for the superiority of our framework. It excels in understanding and answering complex queries, demonstrates advanced reasoning capabilities, and exhibits resilience against the pitfalls of external misinformation. These findings position our framework as a new benchmark in the realms of question-answering and fact-checking, underscoring its comprehensive superiority.

## 3.2 CASE STUDY WITH WEB3 QA APPLICATION

We also apply our framework to develop a QA application in the real-world Web3 domain. Users can ask this QA system up-to-date questions about the Web3 domain. Our system automatically decomposes the user's question into many sub-questions and solves them one by one. In the solving sub-questions process, the system considers injecting knowledge from different sources, such as search engines, existing domain knowledge, and even market databases. Figure 8 illustrates our system's website interface. Despite having a substantial user base and positive user feedback, we rely on expert evaluation to assess our case study and showcase the framework's real-world performance.

**Expert Evaluation.** We design an expert evaluation to assess the quality of explanations and reasoning. Our experts rate explanations on a 1 to 3 scale (with 3 being the best) based on criteria:

- **Coverage**: The explanation and reasoning should cover all essential points important for the fact-checking process.
- **Non-redundancy**: The explanation and reasoning should include only relevant information necessary to understand and fact-check the claim, avoiding unnecessary or repeated details.
- **Readability**: The explanation and reasoning should be clear and easy to read.
- **Overall Quality**: This is a general assessment of the overall quality of the generated explanation and reasoning.

We design an expert evaluation to assess the quality of explanations and reasoning trajectories. Our experts rate these explanations on a 1 to 3 scale (with 3 being the best) based on several criteria: We randomly sample the 100 questions from real users' question history and use React, Self-Ask, and our CoA to answer these questions. Details about expert evaluators selection are shown in Appendix F. Table 6 shows the averaged scores of the expert evaluation. It reveals that CoA outperforms others in expert evaluations, demonstrating its ability to deliver responses that are both more readable and less redundant compared to baseline methods. In summary, these results demonstrate that our framework can get the best performance in the real-world scenario.

## 4 RELATED WORK

We review the literature about prompting methods, agent frameworks, tool learning, and hallucination methods. Owing to page constraints, the contents of tool learning and hallucination methods are relegated to the appendix C.

**Prompting methods.** The key to prompting is to lead LLMs' behavior to follow the instructions. The generic way few-shot prompting [8] enables in-context learning, guiding LLMs to follow instructions and answer questions with only a few examples. CoT [29] and its improved prompting versions [28, 21] try to lead the LLMs to decompose a complex task into a reasoning chain and get better performance. However, they still only support the text information and can not generate the newest information, which is not included in training data.

**Agent frameworks.** Many frameworks aim to expand both the ability and knowledge edges of LLMs. ReAct [33] allows LLMs to interact with external tools to retrieve additional information. Self-ask [18] repeatedly prompts the model to ask follow-up questions to construct the thought process through the search engine. However, these frameworks do not fully harness LLMs' intrinsic knowledge to solve any inner question in the answering process. And they also do not consider the conflicts between LLM-generated content and retrieved information. Search-in-the-Chain [31] relying on the Dense Passage Retrieval (DPR) tries to verify information in the reasoning chain. However, its processing is so complex and sequential that it costs inevitable LLM usages and causes corresponding high latency. Moreover, it still cannot support multimodal data processing. While Chain-of-Knowledge [12] augments LLMs by incorporating grounding information from heterogeneous sources, it highly relies on the fine-tuning of one more LLMs to generate queries sequentially. In addition, it cannot support real-time information. Therefore, we propose a more efficient CoA framework that needs no training cost and supports real-time information. Most importantly, our CoA framework solves sub-questions parallelly, ensuring efficiency.

## 5 CONCLUSIONS AND FUTURE WORK

We introduces the Chain-of-Action (CoA) framework, an innovative approach designed to enhance LLMs capabilities in handling complex tasks, particularly in scenarios where real-time or domain-specific information is crucial. We also propose a efficient verification module utlizing our MRFS to correct the LLM-generated answer by retrieved information. The system successfully demonstrates that detecting the knowledge boundaries of LLMs can improve the efficiency a lot in QA tasks (tokens usage and number of interactions with LLMs). A notable application of CoA is in a Web3 Question Answering product, which demonstrates substantial success in user engagement and satisfaction. It exemplifies the framework's potential in specialized, real-world domains.

## ACKNOWLEDGMENTS

Han Liu is partially supported by NIH R01LM1372201, AbbVie and Dolby. Haozheng Luo is partially supported by the OpenAI Researcher Access Program. Manling Li in partially supported by the Stanford Institute for Human-Centered Artificial Intelligence (HAI), NSF CCRI #2120095, AFOSR YIP FA9550-23-1-0127, ONR MURI N00014-22-1-2740, ONR YIP N00014-24-1-2117, Amazon, and Microsoft. This research utilized computational resources and staff contributions from the Quest High-Performance Computing Facility at Northwestern University, supported by the Office of the Provost, the Office for Research, and Northwestern University Information Technology. The content remains the sole responsibility of the authors and does not necessarily reflect the official views of the funding agencies.

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

# Supplementary Material

## A   FUTURE WORK

Future work includes explorations on the information extraction and analysis on more data modalities, such as vision data [5, 36, 35]. Additionally, we plan to extend CoA to tasks requiring intricate reasoning paths involving recursive or nested logic by implementing an iterative generation mechanism. This approach will involve generating an initial action chain and iteratively refining it based on newly retrieved information, up to a maximum of 10 iterations or until no further retrieval is needed. Such an iterative process aims to minimize irrelevant sub-questions and dynamically adapt the reasoning path, enhancing CoA's applicability to complex scenarios.

Looking forward, we plan to construct a real-time benchmark using back-testing in the Web3 investment market. This benchmark will enable the community to evaluate performance in fast-evolving scenarios and further validate the effectiveness of approaches like CoA. The ultimate goal is to enhance faithfulness and multi-step reasoning for real-world question answering, where comprehensive analysis must sync with external data.

## B   ALGORITHMS

---
**Algorithm 1** Description of Actions Workflow

---
**Initialize:** Actions Chain: AC; Question: Q; LLM Model: M; Query Section: QS; Sub-question: Sub; Guess Answer: A; Faith Score: S; Multi-reference Faith Score: MRFS; Retrieved Results: R; Missing Flag: MF;

**Output:** Final Generated Answer.

**Function** $\text{IR}(Sub_i, A_i, MF_i)$:

    $QS_n = \text{Concat}[Sub_i \mid A_i]$;

    $R = \text{Retrieval}(QS_n)$;

    $MRFS = \arg_k \max S(r_k, A_i)$;

**if** $MF_i ==$ True **then**

    $AC.\text{add}(Sub_i, r_1)$;    // Add Top-1 data

**end if**

**if** $MRFS < T$ **then**

    $AC.\text{correct}(Sub_i, r_k)$;

**end if**

    $AC.\text{add}(Sub_i, r_1)$;

**end Function**

**Function** $\text{MAIN}(Q, M)$:

    $AC = \text{ChainGenerate}(Q, M)$;

**for** each $(Sub_i, A_i, MF_i)$ in AC **do**

    $\text{IR}(Sub_i, A_i, MF_i)$;

**end for**

    **FinalAnswerGenerate**$(AC, M)$;

**return** "Finish";

**end Function**

---

## C   RELATED WORK

**Tool learning.** Recently, tool learning combines the strengths of specialized tools and foundation models to achieve enhanced accuracy and efficiency of problem solving [19]. Toolformer [23] tries to train models to execute APIs for solving problems. Lang2LTL [14] utilizes LLM to ground temporal navigational commands to LTL specifications. However, they mainly focus on specific tasks and domains with delicate algorithm designs. [4] finds that the state-of-the-art methods do not know when they should use tools and when they should simply respond based on their own parametric knowledge. [19] also finds that information conflict between Model Knowledge and Augmented Knowledge is a vital challenge to the accuracy and reliability of model generation and planning.

Hence, our CoA framework is designed to teach LLMs when to request external help and when to solve tasks by themselves with decreasing information conflicts.

**Hallucination methods.** Some work try to solve the hallucination problem by ensemble algorithms [10]. But they are only based on training process without obtaining the real-time information. Retrieval augmentation and verification are the main approaches for mitigating hallucination [7]. Self-Checker [11] comprises many modules including retrieval and veracity prediction for fact-checking by prompting LLMs only. SelfCheckGPT [15] is a black-box zero-resource hallucination verification scheme, which operates by comparing multiple sampled responses and measuring consistency. However, both methods need lots of interactions with LLMs to be inefficient. This drawback motives us to propose the efficient and effective verification module utilizing our MRFS to decrease the hallucination without lots of interactions with LLMs.

## D  PROMPTS

Here are the prompts used in this work:

```
Evaluation Prompt of GPT-4

  Given (question, ground truth answer, LLM-generated answer), you need
to check whether the generated answer contains the ground truth by their
meaning, not individual word only. If correct, the output is 1, otherwise,
0. For example:
[Question]: What should I do when I drink spoiled milk? (A) drink more (B)
drink coffee (C) take some medicine.
[Ground truth]: (C) take some medicine
[Generated answer]: when you drink spoiled milk, you can not drink more or
even drink coffee. You should go to the hospital and check if you need to
take some medicines or not.
[Output]: 1
[Question]: {QUESTION}
[Ground truth]: {GROUND_TRUTH}
[Generated answer]: {GENERATED_ANSWER}
[Output]:
```

## E  MORE EVALUATION DETAILS

Our comparative analysis clearly shows that our approach, which utilizes a global reasoning chain, is more effective than traditional methods that rely on intermediate reasoning or step-by-step question generation and answering. This global perspective allows us to maintain a cohesive view throughout the reasoning process, leading to more accurate and insightful answers. Let's consider a question from the Social QA dataset and compare the responses generated by the CoT method and our CoA approach shown in Figure 7:

**Chain of Thought (CoT) Analysis**: It linearly processes the events: Alex guiding Robin is directly connected to her final meal. It views the action straightforwardly, focusing primarily on the immediate and most obvious context without deeper integration of Alex's roles or broader procedural implications.

**Chain of Action (CoA) Analysis**: (1)Questioning: CoA starts by questioning why Alex would take Robin to the execution chamber, indicating an exploration beyond the immediate action. (2)Investigations: A. 'Work at the jail': CoA explores and affirms Alex's duty to adhere to jail protocols, recognizing that his actions are part of broader job responsibilities. B. 'So Robin can eat': It acknowledges this as the immediate reason but looks further into the procedural context that mandates such actions. C. 'Release her': It dismisses this option based on scenario inconsistency. (3) Final Answer: CoA combines these insights, concluding that Alex's actions are governed by his professional duties to follow protocol, which includes ensuring that condemned prisoners receive their last meal as part of the humane treatment standards.

The answer to the question is B. "so Robin can eat."
Step–by–step analysis:
    1    Alex walked Robin towards the execution chamber – This indicates that Alex is guiding Robin to the location where she will have her last meal before execution.
    2    for her last meal – This further clarifies that the purpose of walking Robin towards the execution chamber is connected to her final meal.
    3    Why did Alex do this?A. "work at the jail" – This is not the reason for Alex walking Robin towards the execution chamber. Alex's job at the jail is not relevant to this specific action.B. "so Robin can eat" – This is the correct reason for Alex's action. The purpose of walking Robin towards the execution chamber is to allow her to have her last meal.C. "Release her" – This does not align with the context provided. Robin is being led towards the execution chamber, so the intention is not to release her.
Therefore, the most reasonable and relevant reason for Alex walking Robin towards the execution chamber for her last meal is B. "so Robin can eat."

{"Question": "Alex walked Robin towards the execution chamber for her last meal. Why did Alex do this? A. 'work at the jail' B. 'so Robin can eat' C. 'Release her'",
  "Chain": [
{"Action": "Questioning","Sub": "Why did Alex walk Robin towards the execution chamber for her last meal?","Guess_answer": "To follow jail protocols for condemned prisoners to have a last meal before execution","missing_flag": "False"},
{"Action": "Investigation","Sub": "work at the jail","Guess_answer": "As a jail guard, Alex's job is to carry out jail procedures for condemned prisoners","missing_flag": "False"},
{"Action": "Investigation","Sub": "so robin can eat","Guess_answer": "To fulfill the prisoner's last request for a final meal before execution","missing_flag": "False"},
{"Action": "Investigation","Sub": "Release her","Guess_answer": "To release Robin from the sentence and grant her clemency","missing_flag": "True"}],
  "Final_answer": "Alex walked Robin towards the execution chamber for her last meal as part of his job as a jail guard to follow the jail protocols for condemned prisoners to have a last meal before execution. Additionally, it fulfills the prisoner's last request for a final meal before execution."}

Figure 7: Responses of baseline and our CoA for the same question. The top and bottom ones are from Chain-of-Thought and our Chain-of-Action, respectively.

*Comparative Insights: Our CoA offers a richer analysis by integrating multiple aspects of the scenario into a comprehensive reasoning chain. This approach addresses the direct question and contextualizes Alex's actions within his duties and the prison's operational protocols, providing a multidimensional understanding. In contrast, CoT tends to a more straightforward, surface-level interpretation. This explanation underscores how CoA's approach provides deeper, more contextually enriched answers compared to CoT, making it particularly effective for complex scenarios requiring a nuanced understanding of actions within their broader social and procedural context.*

# F   CASE STUDY

We first introduce that our expert evaluators were selected from a pool of professionals actively working in the Web3 domain. During the initial stages of product development, we conducted a targeted survey distributed to well-known Web3 practitioners. The survey included 20 questions, with 10 focusing on foundational concepts in Web3 and the remaining 10 being open-ended questions designed to assess their understanding and vision of the Web3 field. Responses were scored, with three senior Web3 investors evaluating the open-ended answers based on their expertise and perspective. From this process, we selected the top 20 candidates with the highest overall scores to serve as our evaluation experts. As an incentive and to ensure continued engagement, these experts were granted free early-stage access to the product. This rigorous selection process was designed to ensure that the evaluators possessed both technical expertise and a nuanced understanding of the Web3 domain. The 20 questions are as follow:

**Conceptual Questions**    The following multiple-choice questions test the foundational knowledge of Web3 practitioners:

1. What is a blockchain?
   - (a) A type of database
   - (b) A distributed ledger technology
   - (c) A centralized system for storing transactions

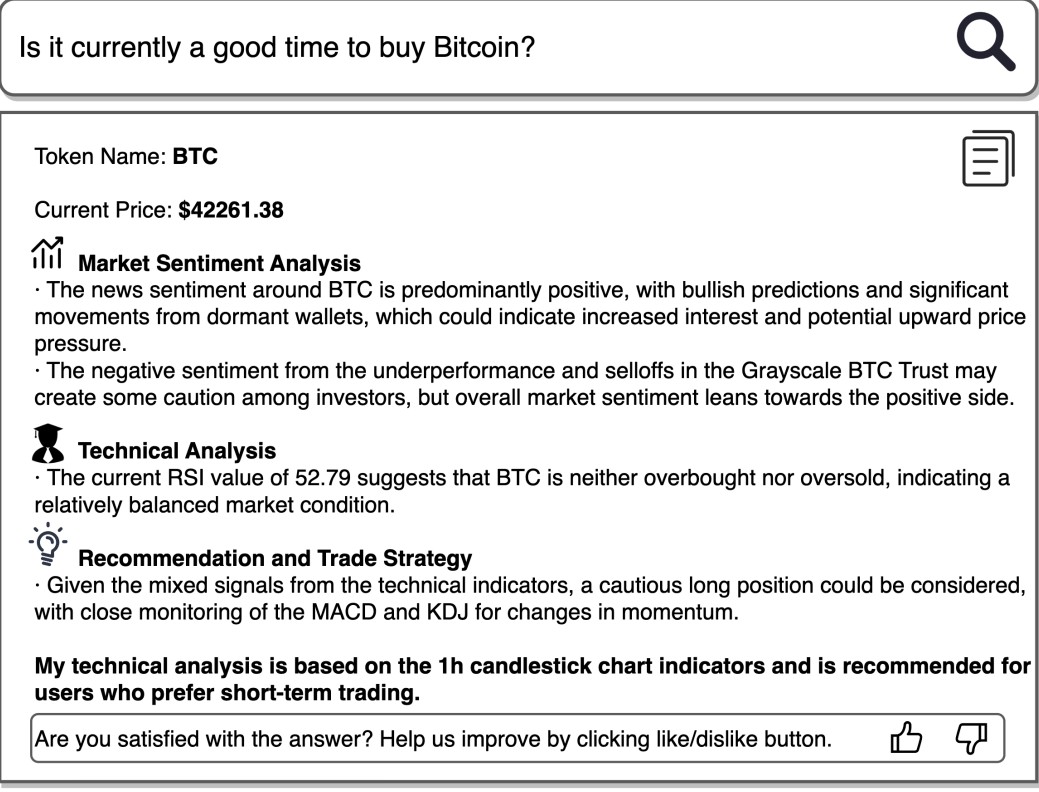

Figure 8: Example of a Web3 QA application interface. In our application, the agent responds to questions and retrieves relevant information for the response.

- (d) None of the above

2. Which of the following best describes smart contracts?
    - (a) Legally binding digital agreements
    - (b) Self-executing programs stored on the blockchain
    - (c) AI-powered decision-making tools
    - (d) Cloud-hosted contracts

3. What is the main purpose of consensus mechanisms in blockchain systems?
    - (a) To ensure security and prevent fraud
    - (b) To store data efficiently
    - (c) To optimize transaction speed
    - (d) To encrypt private keys

4. Which of the following is an example of a Layer 2 scaling solution?
    - (a) Bitcoin
    - (b) Ethereum
    - (c) Polygon
    - (d) IPFS

5. What is the role of tokens in decentralized finance (DeFi)?
    - (a) Representation of digital assets or rights
    - (b) Payment for centralized services
    - (c) Replacement of blockchain miners
    - (d) None of the above

6. What does 'gas fee' refer to in Ethereum transactions?

- (a) The cost of storing data on a centralized server
- (b) The incentive for nodes to validate transactions
- (c) The fee for storing a smart contract
- (d) The cost of staking ETH

7. What is the key difference between Proof-of-Work (PoW) and Proof-of-Stake (PoS)?

- (a) PoW requires miners; PoS uses validators based on stakes
- (b) PoW is faster than PoS
- (c) PoS consumes more energy than PoW
- (d) PoW supports NFTs; PoS does not

8. Which of the following technologies is commonly used to enable Web3 storage?

- (a) IPFS
- (b) Redis
- (c) Firebase
- (d) MySQL

9. What does "interoperability" mean in the context of Web3?

- (a) The ability of a blockchain to scale efficiently
- (b) The compatibility of different blockchain networks
- (c) The execution of transactions in real-time
- (d) The use of AI in blockchain operations

10. What is the primary purpose of decentralized autonomous organizations (DAOs)?

- (a) To govern blockchain systems automatically
- (b) To enable community-driven decision-making without centralized control
- (c) To improve transaction speeds on blockchain networks
- (d) To create legal entities on the blockchain

**Open-Ended Questions**   The following open-ended questions evaluate understanding, creativity, and vision in the Web3 domain:

1. What are the biggest challenges facing blockchain scalability, and how can they be addressed?

2. How a decentralized identity system could revolutionize user privacy and security.

3. What do you think is the most promising application of NFTs beyond art and collectibles?

4. How can blockchain technology improve transparency and trust in traditional financial systems?

5. Explain the potential of Web3 in transforming industries outside of finance, such as healthcare or supply chain management.

6. In your view, what factors determine the success of a Layer 1 blockchain compared to its competitors?

7. How would you design a tokenomics model to incentivize user engagement in a decentralized application (dApp)?

8. What are the trade-offs between decentralization and scalability in current blockchain designs? How would you prioritize them?

9. Discuss the ethical implications of decentralization in Web3 systems. Do you think complete decentralization is desirable? Why or why not?

10. If you were advising a startup launching a DAO, what key considerations would you recommend to ensure its success?

Then, we compare the effectiveness of various web3 case solutions, including Chain-of-Actions, Chain-of-Thought, and the ReAct agent, in addressing complex financial questions.

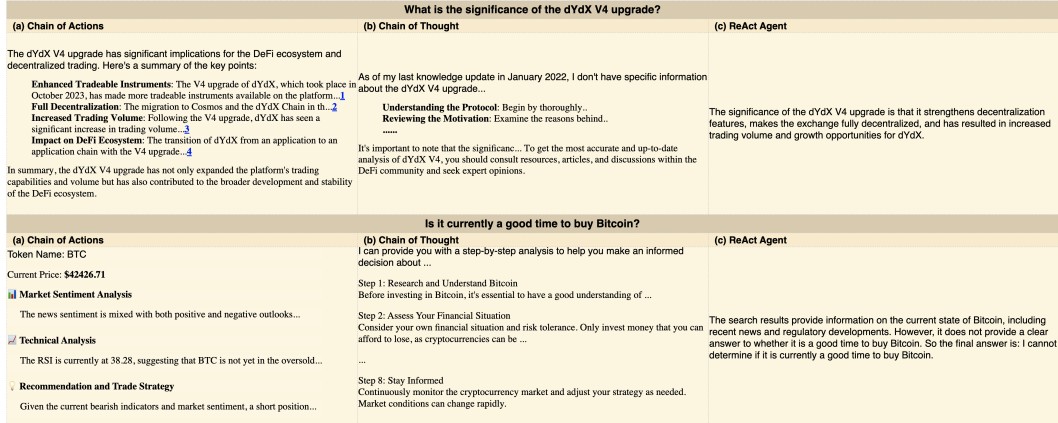

Figure 9: Case studies 1 and 2. Case 1 involves a question that necessitates up-to-date information. Our Chain-of-Actions (CoA) framework efficiently gathers domain knowledge about dYdX and the associated upgrade documentation from the web, subsequently synthesizing this information into a definitive answer. Conversely, the Chain-of-Thought (CoT) approach solely offers guidance on reading the white paper, lacking the capability to access real-time data. The ReAct agent, while successful in locating relevant content via search engines, offers only a broad overview, falling short of providing detailed insights. In case 2, our CoA stands out for providing real-time market price and technical analysis, offering multi-dimensional market insights through a combination of sentiment analysis and technical indicators such as RSI, along with specific trading strategy recommendations, which are invaluable for investors seeking actionable advice. In contrast, the CoT lacks integration of real-time data, focusing more on the decision-making process rather than immediate market actions, while the ReAct, despite aggregating relevant information, falls short by not offering specific guidance and failing to provide a clear conclusion.

Figure 10: Case study 3 and 4. In case 3, our CoA for DOGE provides a comprehensive snapshot by analyzing current market sentiment, which is predominantly positive, and technical indicators like the RSI, which is neutral at 40.66. The recommendation takes into account both these aspects, suggesting caution due to the bearish technical indicators despite the positive sentiment, which is a nuanced approach for traders considering DOGE transactions. However, the CoT and ReAct fall short by not providing specific or updated information, offering general investment advice, and emphasizing personal research and judgment instead of actionable insights. For case 4, the Bitcoin price drop case, our CoA gives a detailed account of factors leading to the price decline, including selling pressure post-ETF approval and temporary profit-taking by investors. It presents a clear narrative of events contributing to the drop, showing a deep dive into the cause-and-effect within the market. The ReAct, on the other hand, acknowledges the lack of clear reasons for the price drop and suggests a variety of possible influences, such as market conditions and investor sentiment, highlighting the complexity and multifaceted nature of cryptocurrency markets but not providing a definitive explanation.

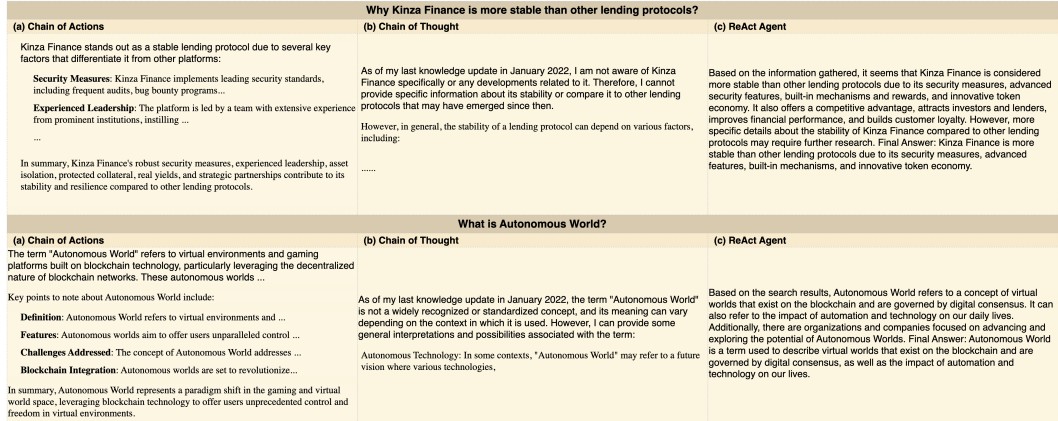

Figure 11: Case study 5 and 6. In case 5, CoA for Kinza Finance highlights its stability as a lending protocol, detailing its robust security measures, experienced leadership, and strategic partnerships. This comprehensive approach emphasizes the unique strengths that contribute to Kinza's stability and resilience in the market. On the contrary, the CoT lacks current updates on Kinza, providing only general factors that affect lending protocol stability, while the ReAct, although it gathers relevant information, only partially addresses why Kinza might be more stable, pointing to security measures and token economy without a thorough analysis. For case 6, in discussing Autonomous World, our CoA provides a clear definition and outlines the impact of such virtual environments on gaming and blockchain technology, citing features, challenges, and the potential for revolutionizing digital interaction. This detailed overview presents a forward-looking view of the integration of blockchain in gaming. However the CoT, with its last update in January 2022, does not offer a precise definition, indicating that the concept can vary widely. Similarly, the ReAct retrieves general information but lacks a focused perspective on the practical implications.

## G    SYSTEM

All experiments are carried out on a cluster, with the exception of the distributed compute node experiment. Each node within the cluster is equipped with 1 NVIDIA GEFORCE RTX 2080 Ti GPUs and 6 8-core Intel XEON Silver 4214 processors running at 2.20GHz. The combined RAM capacity across the cluster nodes amounts to 755GB, and the operating system employed is Ubuntu 18.04.

## H    HYPERPARAMETER

In our experiment, we exclusively require the hyperparameters for LLM, with the exception of Auto-CoT. For the Auto-CoT method, we utilize the KNN model for clustering. Below, we provide a list of all the hyperparameters used in our experiments.

Table 8: Hyperparameter used in the task.

| parameter | values |
|---|---|
| temperature | 0.0 |
| max_length | 1000 |
| top_p | 1.0 |
| n_clusters | 5 |
| retrieval_number | 3 |
| seed | 1 |

