# OpenReview forum: "Chain-of-Action: Faithful and Multimodal Question Answering through Large Language Models"
_ICLR.cc/2025/Conference — ICLR 2025 Poster_

### Official Review · Reviewer_VgSf · 2024-10-31

**Soundness:** 3
**Presentation:** 4
**Contribution:** 3
**Rating:** 5
**Confidence:** 3

**Summary:**

The paper presents the Chain-of-Action (CoA) framework designed to improve large language models' (LLMs) performance in multimodal and retrieval-augmented question-answering (QA). CoA addresses challenges such as hallucination and weak compositional reasoning by decomposing questions into reasoning chains. This method incorporates plug-and-play actions for retrieving diverse data sources and uses a novel multi-reference faith score for verification. Empirical results show CoA outperforms other methods in public benchmarks and real-world applications.

**Strengths:**

- **Innovative Framework**: The CoA's structured decomposition into sub-questions and its use of domain-adaptable plug-and-play actions represent a significant advancement in enhancing the faithfulness and accuracy of LLM responses.
- **Empirical Validation**: Demonstrated strong performance on benchmarks and real-world applications, notably outperforming existing baselines in multimodal QA tasks.
- **Verification Mechanism**: The multi-reference faith score is an effective metric for cross-validating LLM-generated answers against external sources, enhancing reliability.
- **Practical Impact**: Real-world implementation in a Web3 QA system showed increased user engagement and positive feedback, validating the method's applicability.

**Weaknesses:**

- While the CoA approach shows strong empirical performance, its adaptability to more diverse or unstructured data modalities beyond text and tabular data remains to be proven.
- The scalability and efficiency when integrating more complex or real-time data sources require further exploration, especially in scenarios with rapidly changing information.
- The approach, despite its modular design, may face challenges in tasks involving higher-order reasoning or complex multi-step dependencies that are not purely fact-based.

**Questions:**

1. Can the authors provide more details on how the CoA framework could be adapted for tasks involving visual or mixed data modalities?
2. How does the framework handle discrepancies or conflicts when sources provide contradictory information?
3. Are there plans to explore CoA's performance in real-time, fast-evolving information retrieval scenarios where data may change rapidly (e.g., live news events)?
4. Could the use of CoA extend to tasks requiring intricate reasoning paths that involve recursive or nested logic?

---

> ### Author Response · Authors · 2024-11-20
> **Reply to Weakness 1, 2, 3 and Question 1**
>
> Dear Reviewer VgSf:
>
> Thank you for recognizing our work that our structured decomposition and plug-and-play actions enhance LLM response accuracy, with strong empirical validation on benchmarks and real-world Web3 QA applications. And here are our reply to your comments:
>
> ---
>
> Weakness 1:
>
> > While the CoA approach shows strong empirical performance, its adaptability to more diverse modalities remains to be proven.
>
> Answer 1:
>
> Thank you for your comment! Please refer to our response to Question 1, where we address the potential extensions of CoA to more diverse modalities. We hope this provides clarity and addresses your concerns.
>
> ---
>
> Weakness 2:
>
> > The scalability and efficiency when integrating more complex or real-time data sources require further exploration.
>
> Answer 2:
>
> Thank you for your comment! Please refer to our response to Question 3, where we discuss the integration of complex and real-time data sources, including strategies for handling rapidly changing information. We hope this addresses your concerns effectively!
>
> ---
>
> Weakness 3:
>
> > The approach, may face challenges in tasks involving higher-order reasoning or complex multi-step dependencies.
>
> Answer 3:
>
> Thank you for your comment! Please refer to our response to Question 4, where we discuss how CoA addresses tasks requiring intricate reasoning paths and complex multi-step dependencies. We believe the explanation there will provide clarity and address your concerns effectively.
>
> ---
>
> Question 1:
>
> > Can the authors provide more details on how the CoA be adapted for tasks involving visual or mixed data modalities?
>
> Answer 1:
>
> Thank you for your insightful question! The CoA framework currently supports text and tabular data modalities, with its feasibility validated through real-world product deployments. For visual modalities, we have already integrated capabilities in our Web3 QA use case. Specifically, we designed a new action module to process cryptocurrency market trend charts using Vision-Language Models (VLMs). These models extract insights related to sub-questions, such as trend fluctuations or patterns. By combining this visual information with K-line knowledge from local knowledge base, CoA generates more credible answers compared to relying on tabular data.
>
> Also, we experimented with adapting CoA for broader visual and mixed data tasks using advanced vision-language models like Qwen-VL. These models enable CoA to dynamically retrieve and analyze visual information alongside textual data. For instance, when analyzing a cryptocurrency market price trend chart and answering whether a Web3 product is a good investment, CoA retrieves relevant background information, recent news, and supporting data. These inputs are synthesized with Qwen-VL’s visual analysis to provide a comprehensive and well-informed response.
>
> In addition, we have some future plan:
>
> 1. New Actions for Visual Data:
>
> * **Visual-Querying Action:** Similar to web-querying, this action would involve retrieving visual data from sources (e.g., image databases or APIs) relevant to a sub-question. For instance, for an image-based question, the action could retrieve relevant images and pass them to a vision model like Qwen-VL for feature extraction.
> * **Visual-Reasoning Action:** This action would utilize vision-language models such as Qwen-VL to answer sub-questions by interpreting visual inputs in combination with textual context. The retrieved information can then be integrated into the reasoning chain.
> * **Multimodal-Analyzing Action:** For tasks requiring integration of visual and textual data (e.g., charts, annotated images, or multimedia documents), this action can process and align multimodal embeddings using models like Qwen-VL.
>
> 2. Integration with Qwen-VL
>
> * **Reasoning and Retrieval Pipeline:** Qwen-VL’s capability to handle image-text tasks can be seamlessly integrated into the CoA reasoning chain by invoking its API for visual sub-question processing. For example:
>   * If a sub-question asks for identifying objects in an image, the CoA framework can trigger a Qwen-VL-based action.
>   * The embeddings generated by Qwen-VL can serve as inputs to subsequent actions for cross-modal reasoning.
> * **Multi-Reference Faith Score (MRFS) for Visual Tasks:** The MRFS metric can be extended to verify the alignment between retrieved visual data and LLM-generated responses, ensuring faithfulness in multimodal tasks.
>
> 3. Future Vision:
>
> * The modularity of CoA allows the addition of other advanced vision-language models, including task-specific fine-tuned variants, to further expand capabilities.
> * Future work could explore fine-tuning the interaction protocols between textual reasoning and visual data interpretation.
>
> We thank the reviewer for their constructive proposal, and we believe incorporating Qwen-VL and other vision-language tools into CoA will significantly broaden its applicability and strengthen its contributions to multimodal and retrieval-augmented QA
>
> ---

---

> ### Author Response · Authors · 2024-11-20
> **Reply to Question 2, 3**
>
> ---
>
> Question 2:
>
> > How does CoA handle discrepancies or conflicts when sources provide contradictory information?
>
> Answer 2:
>
> Thank you for this excellent question! Handling discrepancies or conflicts when sources provide contradictory information is indeed a critical challenge. In our current framework, when performing web searches, we retrieve the top *k* most similar results and pass them all to the LLM, prompting it to select the most reliable answer. This approach has the advantage of saving time, as it allows us to obtain a single result in one step. However, it can introduce instability due to the LLM's reliance on potentially conflicting sources.
>
> We have also considered an alternative approach where each page is summarized individually into potential answers, followed by a voting mechanism to identify the most common or consistent information. While this method can improve accuracy by leveraging consensus, it requires multiple LLM calls, significantly increasing latency. As such, we have made a trade-off between time and accuracy in our current design.
>
> Looking ahead, we plan to explore faster and more effective methods to resolve conflicts among external data sources. For example, assigning weights to different sources based on their credibility or using additional retrieval rounds for fact-checking could help improve both reliability and efficiency. These directions will further enhance CoA's ability to handle contradictory information in a robust and timely manner. Thank you again for highlighting this important aspect!
>
> ---
>
> Question 3:
>
> > Are there plans to explore CoA's performance in real-time, fast-evolving information retrieval scenarios where data may change rapidly (e.g., live news events)?
>
> Answer 3:
>
> Thank you for the excellent question! Yes, as highlighted in our paper, the Web3 QA use case is a prime example of a real-time, fast-evolving scenario. The Web3 field is rapidly growing, with new products and concepts emerging daily, often tied to specific cryptocurrencies. In our system, investment advice is the most frequently queried category, accounting for approximately 68% of total queries. To address these queries effectively, we need to quickly identify the mentioned products and retrieve relevant information via web search and market data for the LLM to process. User feedback suggests that the current results are satisfactory for most users. However, investment advice inherently involves subjective factors and unpredictability, so users typically treat the LLM’s suggestions as references rather than absolute decisions. In high-frequency real-world applications, we optimize the retrieval process through engineering efforts like parallelization and fuzzy search to reduce latency. For fairness in our experiments, we used a standardized retriever across baselines. However, in practical scenarios, each task—such as vector database management, vector search, or web search—has dedicated teams working to improve their performance. This means that CoA can further benefit from advancements in any of these areas, leading to a synergistic evolution. **Additionally, we have included experiments in Table 3 of revised version showing the LLM’s input and output token usage, as well as the overall average latency.** These results demonstrate that CoA outperforms other RAG-based frameworks by leveraging the LLM’s parametric knowledge to significantly reduce overall costs and latency. Looking forward, we plan to construct a real-time benchmark using back-testing in the Web3 investment market. This benchmark will enable the community to evaluate performance in fast-evolving scenarios and further validate the effectiveness of approaches like CoA. **We have added it in the Appendix as our future work.** Thank you again for your thoughtful question!

---

> ### Author Response · Authors · 2024-11-20
> **Reply to Question 4:**
>
> ---
>
> Question 4:
>
> > Could the use of CoA extend to tasks requiring intricate reasoning paths that involve recursive or nested logic?
>
> Answer 4:
>
> Thank you for the insightful question! Yes, our current CoA framework is designed to leverage the LLM's parametric knowledge and our MRFS verification mechanism to significantly reduce redundant external processing, costs, and latency. While CoA already demonstrates strong performance on complex tasks, such as multi-turn open-domain datasets like QReCC and long-form datasets like ASQA, we have also devised a solution for handling even more intricate reasoning paths: an iterative generation mechanism.
>
> This approach involves generating an initial action chain and iteratively refining it. If any sub-node requires external retrieval for correction or imputation, the completed chain is fed back as input to regenerate an updated chain. This process continues for up to 10 iterations or until no sub-node requires retrieval, dynamically adapting the reasoning path based on the latest information. By doing so, we can minimize the generation of irrelevant sub-questions that may occur in a single-generation process. However, this iterative approach requires additional processing time, which is why in our current paper, we limit CoA to a single generation for efficiency. Our findings indicate that even with this limitation, CoA's performance is satisfactory for most tasks. **Additionally, we have included this iterative mechanism in the appendix of the revised version as a potential direction for future work.** In the future, we aim to explore how to achieve optimal performance with minimal time overhead, potentially making iterative refinements more practical for more complicated applications. Thank you again for your thoughtful feedback!

---

> > ### Author Response · Authors · 2024-11-22
> > **Follow-Up on Rebuttal Phase Feedback**
> >
> > Dear Reviewer,
> >
> > Thank you once again for reviewing our paper and providing such valuable and constructive feedback. We have carefully studied your suggestions and made several revisions, adding extensive experimental details to enhance the paper’s clarity, depth, and contributions. Your insightful comments have been instrumental in guiding these improvements.
> >
> > We sincerely hope that if you have any further questions or concerns, you will not hesitate to let us know. We are more than willing to provide additional clarifications and supporting materials to ensure the paper meets the highest standard. Your insights are crucial for refining our research and ensuring its relevance and impact.
> >
> > Additionally, we kindly hope that these updates and clarifications will encourage you to reconsider your evaluation, as they directly address your constructive feedback. Should you have any additional queries or reservations, please feel free to contact us at any time. We are fully committed to addressing all concerns to your satisfaction.
> >
> > Thank you once again for your invaluable time and support.
> >
> > Best regards,
> >
> > Authors

---

### Official Review · Reviewer_wYMh · 2024-11-03

**Soundness:** 3
**Presentation:** 3
**Contribution:** 3
**Rating:** 8
**Confidence:** 3

**Summary:**

The authors propose a new QA retrieval mechanism called Chain of Action(CoA). When a question is asked to an LLM, there is a prompt which generates a list of actions the LLM needs to take first to effectively answer the questions. They introduce a Plug and Play approach where in case of Multimodal LLMs, the actions taken can be integrated into the application. The actions can be web query or data analysis. The paper integrates 3 such actions. The LLM then performs each of the individual action generated and then there is another query which combines information from all the actions. The LLM then gives an answer based on the newly injected information

**Strengths:**

1. The authors utilize newer multimodal LLM abilities to perform actions such as web query and data analysis. The authors come up with a new QA mechanism for LLMs which uses the actions The method is called Chain of Action(CoA).
2. The authors demonstrate that this method significantly outperforms other reasoning and QA methods on many QA datasets.
3. The improvement of using actions over thoughts does seem to be the natural way of solving a question. This approach has significant potential for improving QA capabilities of LLMs.

**Weaknesses:**

1. Based on the number of actions to be taken and what kind of "plug" is used for the action, the time taken to finish all actions and send out an answer might become significant. It would have been good to see the study on latency(eg. average response time) of the system because of the new method.
2. It would be helpful to conduct an ablation study when you remove specific action types to isolate their impact on performance. This would provide clearer insights on how much this method relies on additional capabilities.
3. Comparing CoA with and without the ability to perform additional "plugs" across different types of questions can be useful in understanding the impact of this method.

**Questions:**

1. Can you elaborate on the key differences between "thoughts" in CoT and "actions" in CoA? How does this change improve the overall performance? It would also be helpful if you can discuss the limitations and trade-offs between them.
2. If the system doesn't have the ability to add additional actions like web query, does CoA still perform better than CoT.
3. Does CoA add significant latency to QA process?

---

> ### Author Response · Authors · 2024-11-20
> **Reply to Weakness 1, 2, 3 and Question 3**
>
> Dear Reviewer wYMh,
>
> Thanks so much for your commending our innovative Chain of Action (CoA) mechanism, which leverages multimodal LLM capabilities to integrate actions like web querying and data analysis for QA tasks, noting its significant performance improvements over traditional reasoning methods and its potential to advance QA capabilities. And here are our reply to your comments:
>
> ---
>
> Weakness 1:
> >Based on the number of actions to be taken and what kind of "plug" is used for the action, the time taken to finish all actions and send out an answer might become significant....
>
> Answer 1:
>
> Thank you for raising this important point. We completely agree that incorporating a study on latency is essential to highlight the advantages of our method. **In the revised version, we have updated Table 3 to include detailed input and output token consumption, as well as the average latency for all baselines.** These results demonstrate that our method effectively leverages the parametric knowledge of large models and employs MRFS for verification, significantly reducing the usage of LLM tokens and minimizing unnecessary retrievals. For fairness and validity, we used the same vector-based retrieval process for similar passages as the baselines. In our Web3 QA case study, we further optimized the retrieval process through engineering efforts like parallel processing, which greatly reduced the latency for each response. This optimization ensures that our method meets the real-time demands of users seeking answers about rapidly changing markets. We hope these additions address your concerns and provide a clearer understanding of our system's efficiency. Thank you again for your thoughtful feedback!
>
> ---
>
> Weakness 2:
> >It would be helpful to conduct an ablation study when you remove specific action types to ...
>
> Answer 2:
>
> Thank you for your helpful suggestion. **We have addressed this in the revised version by adding more results to Table 2, specifically showing the performance impact when Action 1 (web search) and Action 2 (local knowledge base search) are removed**. From the results, it is evident that the improvement brought by web search is more significant compared to local knowledge base search, especially for non-factual and open-ended QA scenarios. These insights further clarify the relative contribution of each action type to the overall performance. We appreciate your feedback in helping us enhance the analysis!
>
> ---
>
> Weakness 3:
>
> >Comparing CoA with and without the ability to perform additional "plugs" across different types of questions ...
>
> Answer 3:
>
> Thank you very much for this insightful suggestion. **We have conducted further analysis to compare the performance of CoA with and without retrieval across different types of questions.** Our findings show that CoA's external "plugs" significantly enhance performance in complex scenarios, such as open-domain questions requiring cross-domain knowledge transitions (e.g., QReCC) and long-form questions that demand detailed, structured answers. In contrast, the improvements are relatively smaller for simpler, commonsense questions, where the LLM's parametric knowledge is usually sufficient. This analysis highlights CoA's strength in addressing complex problems and its ability to effectively extend the model's capabilities beyond its inherent limitations. We appreciate your feedback, which helped us deepen understanding of our method's impact!
>
> ---
>
> Question 3:
> >Does CoA add significant latency to QA process?
>
> Answer 3:
>
> Thank you for your thoughtful question. The impact of CoA on latency depends on the context. Compared to basic baselines that rely solely on the capabilities of large language models (LLMs) for QA, CoA does introduce additional latency due to external data retrieval and processing. However, existing research has consistently shown that relying solely on LLMs is insufficient for addressing many real-world problems, such as small-scale events or real-time information. Therefore, our primary comparison is against RAG-based baselines. **As shown in Table 3 of revised version, our experiments demonstrate that CoA significantly reduces various metrics compared to other RAG baselines, including overall latency, the number of LLM calls, and the token costs for LLM input and output.** These results highlight that by simply detecting the knowledge boundaries of LLMs, leveraging their parametric knowledge, and verifying with our MRFS, CoA can substantially reduce the need for external data retrieval, making the overall process more efficient and cost-effective. We hope this addresses your concern and provides clarity on the advantages of our approach. Thank you again for your valuable feedback!

---

> ### Author Response · Authors · 2024-11-20
> **Reply to Question 1, 2**
>
> ---
>
> Questions 1:
> >Can you elaborate on the key differences between "thoughts" in CoT and "actions" in CoA? How does this change improve the overall performance? It would also be helpful if you can discuss the limitations and trade-offs between them.
>
> Answer 1:
>
> Thank you for your thoughtful question. The key difference between "thoughts" in CoT and "actions" in CoA lies in their approach to handling complex problems and their reliance on LLM capabilities. The core of CoT is to decompose a complex problem into sequential steps (thoughts) that a transformer-based model can solve within its theoretical capacity. Each "thought" represents one step, and the process of generating them is constrained entirely by the LLM's inherent abilities. This approach relies solely on the LLM's parametric knowledge to progressively handle each step until arriving at a final answer. In contrast, CoA  extends this framework by introducing external tools that the LLM can utilize. CoA enables the model to consider the availability of retrieval tools when decomposing a complex problem. Instead of being limited to internal reasoning, CoA first addresses sub-questions that fall within the LLM's knowledge boundaries and delegates sub-questions beyond its scope to specific external tools. Each "action" in CoA is a structured unit comprising the decomposed sub-question, the assigned tool, and a flag indicating whether external help is required. This transition from "thoughts" to "actions" introduces a more nuanced and multidimensional analysis process compared to the straightforward decomposition in CoT. By integrating different dimensions of reasoning and external knowledge retrieval, CoA enhances the ability to address real-world scenarios that often require more context and dynamic information than CoT can provide.
>
> Improvement in performance:
>
> 	This shift improves overall performance in scenarios where LLMs alone are insufficient, such as those requiring real-time information or highly specialized external knowledge. By leveraging tools dynamically, CoA reduces the dependency on LLMs to guess or approximate answers, resulting in higher accuracy and richer, more contextually grounded responses.
>
> Limitations and Trade-offs:
>
> 	Limitations of CoA:
> 	Latency: Incorporating external tools adds latency, as retrieval and processing require additional time.
> 	Tool Dependency: The performance depends on the quality and reliability of the external tools.
> 	Implementation Complexity: Designing actions and ensuring smooth integration with tools can be more complex than CoT.
>
> 	Trade-offs Compared to CoT:
> 	Simplicity vs. Capability: CoT is simpler and faster as it relies only on the LLM but is limited by its parametric knowledge. CoA sacrifices simplicity for enhanced capability by integrating external tools.
> 	Scalability: While CoT scales efficiently within the LLM, CoA requires careful scaling and optimization of the retrieval process for practical applications.
> We hope this clarifies the distinctions and trade-offs between CoT and CoA and how the latter improves performance while addressing real-world challenges. Thank you again for your insightful question!
>
> ---
>
> Question 2:
> >If the system doesn't have the ability to add additional actions like web query, does CoA still perform better than CoT.
>
> Answer 2:
>
> Thank you for the excellent question. Yes, even without additional actions like web query, CoA still performs better than CoT. **In Table 2, we conducted an initial evaluation by comparing the CoA without Retrieval version against baselines that rely solely on the LLM's inherent capabilities**. The results show that CoA still outperforms all baselines. To further investigate the reasons behind this, we selected a representative case study and provided a **detailed analysis in Appendix Section E**. This comparison highlights that CoA offers a richer analysis by integrating multiple aspects of the scenario into a comprehensive reasoning chain. Unlike CoT, which often produces straightforward, surface-level interpretations, CoA contextualizes responses within broader social and procedural frameworks, addressing both the direct question and its underlying implications. This insight demonstrates how CoA provides deeper, more contextually enriched answers compared to CoT, making it particularly effective for complex scenarios requiring nuanced understanding. We hope this addresses your question thoroughly, and we appreciate your thoughtful input!

---

> > ### Author Response · Authors · 2024-11-22
> > **Follow-Up on Rebuttal Phase Feedback**
> >
> > Dear Reviewer,
> >
> > Thank you once again for reviewing our paper and providing such valuable and constructive feedback. We have carefully studied your suggestions and made several revisions, adding extensive experimental details to enhance the paper’s clarity, depth, and contributions. Your insightful comments have been instrumental in guiding these improvements.
> >
> > We sincerely hope that if you have any further questions or concerns, you will not hesitate to let us know. We are more than willing to provide additional clarifications and supporting materials to ensure the paper meets the highest standard. Your insights are crucial for refining our research and ensuring its relevance and impact.
> >
> > Additionally, we kindly hope that these updates and clarifications will encourage you to reconsider your evaluation, as they directly address your constructive feedback. Should you have any additional queries or reservations, please feel free to contact us at any time. We are fully committed to addressing all concerns to your satisfaction.
> >
> > Thank you once again for your invaluable time and support.
> >
> > Best regards,
> >
> > Authors

---

> ### Comment · Reviewer_wYMh · 2024-12-03
>
> Thank you for the detailed response. I am satisfied the additional studies performed. I would recommend adding details about  experiments mentioned in "Answer 3" to the paper. This study is crucial to understand where the real gains are coming from.

---

> > ### Author Response · Authors · 2024-12-03
> >
> > Thank you for your valuable feedback and for highlighting the importance of the experiments mentioned in "Answer 3." We are pleased to inform you that we have already incorporated these details into the paper to provide a more comprehensive understanding of where the real gains are coming from. Specifically, the additional content is included in Section 3.1.1, which discusses the performance of CoA with and without retrieval across various question types:
> >
> > ***Table 2 also shows that CoA's external "plugs" significantly enhance performance on average 15.3% in complex scenarios, such as open-domain questions requiring cross-domain knowledge transitions (e.g., QReCC) and long-form questions that demand detailed, structured answers. In contrast, the improvements are relatively smaller for simpler, commonsense questions (7.2% on average), where the LLM's parametric knowledge is usually sufficient. This result highlights CoA's strength in addressing complex problems and its ability to extend the model's capabilities effectively.***
> >
> > Unfortunately, as the rebuttal phase does not allow for uploading a revised version of the paper, we are unable to share the updated version at this time. We hope for your kind understanding, and we assure you that the updated version will be available in the next revision.
> >
> > Thank you again for your insightful comments and support.
> >
> > Best regards,
> >
> > Authors

---

### Official Review · Reviewer_mjpZ · 2024-11-03

**Soundness:** 2
**Presentation:** 1
**Contribution:** 2
**Rating:** 8
**Confidence:** 4

**Summary:**

The paper introduces the Chain-of-Action (CoA) framework, a novel approach to multimodal and retrieval-augmented question answering that enhances the faithfulness and reasoning quality of large language models (LLMs). CoA addresses key challenges in QA, such as unfaithful responses and weak reasoning, by decomposing questions into a series of reasoning steps or actions that systematically retrieve and verify information from various sources. The framework introduces three "Plug-and-Play" actions—web querying, knowledge encoding, and data analyzing—that support multimodal data integration. Additionally, a multi-reference faith score (MRFS) is proposed to resolve inconsistencies and improve response accuracy. Experimental results demonstrate CoA’s effectiveness in handling complex questions across QA benchmarks and in real-world applications, particularly in the Web3 domain.

**Strengths:**

1. This study introduces a framework embodying the divide-and-conquer approach, effectively breaking down complex tasks into manageable components that are tackled sequentially. This structure enhances the model's ability to handle multifaceted queries with improved precision.

2. The empirical results demonstrate notable improvements in both performance and efficiency, as reflected in reduced API calls and token usage compared to prior methods. These gains underscore the framework’s effectiveness and potential for cost-saving in real-world applications.

3. The introduction of the multi-reference faith score (MRFS) is a contribution, which effectively identifies and mitigates information conflicts, and improves answer reliability and trustworthiness in real-time applications.

**Weaknesses:**

1. The paper’s primary weakness lies in how it presents its key concepts and narrative. Many claims, such as "multimodal," "plug-and-play," and "action-based" elements, lack direct evidence or clear definitions, making it challenging to follow the core contributions. Though the pipeline is straightforward, understanding the study's actual workflow is hindered by (1) inaccurate terminology, (2) loosely connected methodology descriptions, and (3) a mix of abstract workflows and technical details.

2. Certain terms are uncommon or seem misapplied, which leads to confusion. For example, terms like "multimodal" (when referring to text and tabular data), "chain-of-action" (more of a "chain-of-data-collection-and-verification"), "actions design" (data collection), "actions workflow" (data verification), "node" (sub-question), and "knowledge boundary" (what a model actually knows) lack clarity and could benefit from more precise definitions or alternatives.

3. Question decomposition appears critical to this framework, yet there is limited discussion on decomposition strategies or comparisons with existing baselines. Further elaboration here would strengthen the paper's contributions.

4. The "plug-and-play" feature is presented as a low-cost prompting strategy; however, integrating retrieval for each data type (e.g., web, internal knowledge) may not be straightforward. It may be worth reconsidering or refining this claim to better reflect its implementation complexity.

5. The paper’s claim of multimodal data handling is unclear. If the input consists of real-time information, domain knowledge, and tabular data, it may be more accurately described as handling heterogeneous data rather than multimodal data. Additionally, if tabular data is linearized as text for LLM input, the fundamental multimodal claim weakens.

6. The study does not include ablations to show the specific contribution of tabular data. Providing such analyses could clarify its impact on the framework's performance.

7. Section 3.2 mentions expert evaluation on a 1 to 3 scale based on three criteria, but it lacks details on the expert recruitment process, qualifications, and any inter-rater reliability metrics. Adding these details would increase the transparency and credibility of the evaluation process.

**Questions:**

1. Could you clarify what “imputation” refers to in Table 2? Are there results available for CoA without MRFS, and what does “w/ ROUGE” mean? My understanding was that ROUGE is used only in ASQA.

2. In Table 3, could you provide separate statistics for input and output tokens, as well as the average token usage per action? This would help readers better understand the specific cost details.

3. Could you elaborate on what is meant by the term “knowledge boundary”?

4. Are the results of the Chain-of-Action framework directly comparable to previous studies? I noticed that this study used GPT-4, while DSP and SearchChain relied on older-generation LLMs (text-davinci-002 and gpt-3.5-turbo, respectively).

5. Would it be fair and perhaps clearer to rename Sections 2.2.1 and 2.2.2 as "Data Collection" and "Data Verification," instead of “Actions Design” and “Actions Workflow”? These alternative terms seem easier to understand and align well with the content of the corresponding subsections.

---

> ### Author Response · Authors · 2024-11-20
> **Reply to Weakness 1,2,3,4,5**
>
> Dear Reviewer mjpZ:
>
> We sincerely thank the reviewer for recognizing our framework's contributions to improving precision, efficiency, and cost-saving potential, as well as highlighting the value of the multi-reference faith score (MRFS) in enhancing answer reliability. Your thoughtful feedback is greatly appreciated! And here are our reply to your comments:
>
> ---
>
> Weakness 1, 2, 5:
>
> > Though the pipeline is straightforward, understanding the study's actual workflow is hindered by (1) inaccurate terminology, (2) loosely connected methodology descriptions, and (3) a mix of abstract workflows and technical details. Certain terms are uncommon or seem misapplied, which leads to confusion. For example, terms like "multimodal" (when referring to text and tabular data), "chain-of-action" (more of a "chain-of-data-collection-and-verification"), "actions design" (data collection), "actions workflow" (data verification), "node" (sub-question), and "knowledge boundary" (what a model actually knows) lack clarity and could benefit from more precise definitions or alternatives.
>
> Answer 1, 2, 5:
>
> Thank you for your valuable feedback. **In the revised version of our paper, we have carefully addressed the issues you highlighted regarding terminology and clarity.** Specifically, we have refined the definitions and replaced terms like "multimodal", "actions design", and "actions workflow" with more precise alternatives to reduce ambiguity and better align with their intended meanings. For example, we now describe the input as "heterogeneous data" instead of "multimodal data,". We also added a clear definition of knowledge boundary in the answer to Question 3. These changes aim to improve clarity and ensure the terminology is both accurate and intuitive for readers. We believe these updates will effectively address the concerns you raised and enhance the overall readability and precision of the paper.
>
> ---
>
> Weakness 3:
>
> > Question decomposition appears critical to this framework, yet there is limited discussion on decomposition strategies or comparisons with existing baselines. Further elaboration here would strengthen the paper's contributions.
>
> Answer 3:
>
> Thank you for your insightful suggestion. We agree that further elaboration on decomposition strategies strengthens the paper. In Table 2, we compare CoA without actions against other decomposition baselines, emphasizing the differences in reasoning and decomposition methods. Additionally, Appendix D provides a case study comparing CoA and CoT to further clarify these distinctions. To validate the correctness and relevance of decomposed sub-queries, we evaluated 50 questions from the Social QA dataset, yielding 96% correctness and 98% relevance in the CoA, compared to 92% and 96% for CoT. These results highlight strong alignment between sub-queries, decisions, and outcomes. We are also exploring automated evaluation methods for relevance and correctness to improve scalability. **These updates have been included in the revised version**, and we hope they address your concerns. Thank you again for your valuable feedback!
>
> ---
>
> Weakness 4:
>
> > The "plug-and-play" feature is presented as a low-cost prompting strategy; however, integrating retrieval for each data type (e.g., web, internal knowledge) may not be straightforward. It may be worth reconsidering or refining this claim to better reflect its implementation complexity.
>
> Answer 4:
>
> Thank you for your valuable feedback. We agree that integrating retrieval for different data types is indeed not straightforward, and we appreciate the opportunity to clarify this point. **In the introduction section of the revised version of the paper, we have refined the claim regarding the "plug-and-play" feature to provide a more accurate description**. Specifically, we now state that: “*The term 'plug-and-play' refers to the ability to freely add or remove pre-designed actions, such as the three different actions implemented in our work. However, for any new action to be integrated in the future, careful design and adjustment will be required to ensure compatibility with the framework's input and output formats.*” We believe this revision more accurately reflects the complexity of implementation while maintaining the core idea of flexibility in extending the framework. Thank you again for your thoughtful suggestion, which has helped improve the clarity of our work.

---

> ### Author Response · Authors · 2024-11-20
> **Reply to Weakness 6, 7 and Question 1**
>
> Weakness 6
>
> > The study does not include ablations to show the specific contribution of tabular data. Providing such analyses could clarify its impact on the framework's performance.
>
> Answer 6:
>
> Thank you for your thoughtful suggestion. We agree that it is important to provide a comparison to clarify the specific contribution of tabular data. As described in Section 2.2.1 - Action 3, our tabular data action is used exclusively in the Web3 case to retrieve market data. **Based on your feedback, we revisited our experiments and consulted the Web3 experts we worked with earlier to evaluate the performance of the framework in scenarios where tabular data is not available for support**. **The results of this analysis have been included in the revised version, specifically in Table 6, as part of an ablation study.** From the updated results, we observe a significant decline in the coverage and overall quality metrics without the inclusion of real-time market prices, highlighting the lack of truly useful references. However, the non-redundancy and readability metrics show negligible differences. We hope this additional analysis provides clarity and addresses your concern.
>
> ---
>
> Weakness 7:
>
> > Section 3.2 mentions expert evaluation on a 1 to 3 scale based on three criteribut it lacks details on the expert recruitment process, qualifications, and any inter-rater reliability metrics. Adding these details would increase the transparency and credibility of the evaluation process.
>
> Answer 7:
>
> Thank you for pointing out the need for additional details regarding the expert evaluation process. We appreciate the opportunity to provide further clarity. Our expert evaluators were selected from a pool of professionals actively working in the Web3 domain. During the initial stages of product development, we conducted a targeted survey distributed to well-known Web3 practitioners. The survey included 20 questions, with 10 focusing on foundational concepts in Web3 and the remaining 10 being open-ended questions designed to assess their understanding and vision of the Web3 field. Responses were scored, with three senior Web3 investors evaluating the open-ended answers based on their expertise and perspective. From this process, we selected the top 20 candidates with the highest overall scores to serve as our evaluation experts. As an incentive and to ensure continued engagement, these experts were granted free early-stage access to the product. This rigorous selection process was designed to ensure that the evaluators possessed both technical expertise and a nuanced understanding of the Web3 domain. Additionally, **we have included a detailed description of this process and 20 questions in the revised version's Appendix Section F**, as we believe this addition strengthens the reliability and transparency of the paper. Thank you again for your thoughtful feedback.
>
> ---
>
> Question 1:
>
> > Could you clarify what “imputation” refers to in Table 2? Are there results available for CoA without MRFS, and what does “w/ ROUGE” mean? My understanding was that ROUGE is used only in ASQA.
>
> Answer 1:
>
> 1.1. Clarify “imputation”:
>
> The term "imputation" in Table 2 refers to the process when llm encounters a sub-question that it cannot answer (and leaves the initial answer blank), we utilize retrieved information to provide an answer for that sub-question.  **We have clarified the meaning in the Section 3.1-Baselines of the revised version.**
>
> 1.2. Results of CoA without MRFS
>
> We are sorry to confuse you. The MRFS only exists in the verification part. So, the CoA without MRFS is equal to the CoA without verification, whose results are already listed in Table 2. **We have changed the name of CoA-MRFS to CoA(MRFS in verification) in Table 2 of the revised version.**
>
> 1.3. Meaning of “w/ROUGE”
>
> We apologize for any confusion regarding this term. The ROUGE in Table 2 is different from the way ROUGE-L is used in the ASQA dataset. In our work, the verification metric MRFS (in Section 2.2.2) was inspired by the original ROUGE. Therefore, we included "verification with ROUGE" in Table 2 as a baseline for comparison against "verification with MRFS." It highlights that our MRFS yields better verification than the original ROUGE. On the other hand, in the ASQA dataset, ROUGE-L is used specifically to evaluate long-form QA responses, which serves a different purpose than the ROUGE in Table 2. Notably, ROUGE-L focuses on the Longest Common Subsequence (LCS), making it particularly suitable for evaluating the fluency and coherence of long-form answers, which often have more diverse phrasing and structure compared to shorter responses. **We have clarified the difference between ROUGH and ROUGH-L in the ASQA dataset in Section 3.1-Metrics of the revised version.**

---

> ### Author Response · Authors · 2024-11-20
> **Reply to Question 2, 3, 4, 5**
>
> ---
>
> Question 2:
>
> > In Table 3, could you provide separate statistics for input and output tokens, as well as the average token usage per action? This would help readers better understand the specific cost details.
>
> Answer 2:
>
> Thank you for your valuable suggestion! **We have added the requested results to the revised version and updated Table 3 to include separate statistics for input and output tokens.** Additionally, for actions like web query and database search, we found that the average token usage ratio is approximately 7:3. **We have also included the corresponding average time cost to provide a more comprehensive understanding of the cost details.** We hope these updates address your concerns and improve the clarity of our paper. Thank you again for your helpful feedback!
>
> ---
>
> Question 3:
>
> > Could you elaborate on what is meant by the term “knowledge boundary”?
>
> Answer 3:
>
> Thank you very much for your question. By "knowledge boundary," we are referring to the parametric knowledge that the model has already acquired during training on a high-quality dataset. Given that LLMs with extensive parameters have undergone costly training, leveraging the comprehensive knowledge boundary that the LLM has already mastered can significantly reduce unnecessary retrieval efforts—particularly during the filtering and summarization phases after coarse retrieval—in a RAG-based framework. In other words, it helps decrease the LLM interaction frequency and token usage during the QA task. **We have clarified the meaning more clearly in the introduction section.**
>
> ---
>
> Question 4:
>
> > Are the results of the Chain-of-Action framework directly comparable to previous studies? I noticed that this study used GPT-4, while DSP and SearchChain relied on older-generation LLMs (text-davinci-002 and gpt-3.5-turbo, respectively).
>
> Answer 4:
>
> Thank you very much for your question, and I apologize for any confusion. As mentioned in Section 3.1 under "Implementation," the reference to GPT-4 in our paper only belongs to the evaluation process, where we use it to assess whether the answers generated by different baselines, including our proposed CoA framework, align with the ground truth.(see Appendix C for the evaluation prompt details). For the answer generation itself, all baselines, including our CoA, utilize the same backbone model, gpt-3.5-turbo. This ensures a fair and trustworthy comparison across all experiments presented in the study. We greatly appreciate your pointing this out, and **we have clarified this more clearly in the Implementation part as well**.
>
> ---
>
> Question 5:
>
> > Would it be fair and perhaps clearer to rename Sections 2.2.1 and 2.2.2 as "Data Collection" and "Data Verification," instead of “Actions Design” and “Actions Workflow”? These alternative terms seem easier to understand and align well with the content of the corresponding subsections.
>
> Answer 5: Thank you very much for your suggestion. After careful consideration, we agree that the content under "Actions Workflow," including Answering Verification and Missing Detection, can indeed be summarized as "Data Verification." Using "Data Collection" and "Data Verification" as the new titles would make these sections easier to understand. **We have updated the corresponding section titles accordingly.**

---

> > ### Author Response · Authors · 2024-11-22
> > **Follow-Up on Rebuttal Phase Feedback**
> >
> > Dear Reviewer,
> >
> > Thank you once again for reviewing our paper and providing such valuable and constructive feedback. We have carefully studied your suggestions and made several revisions, adding extensive experimental details to enhance the paper’s clarity, depth, and contributions. Your insightful comments have been instrumental in guiding these improvements.
> >
> > We sincerely hope that if you have any further questions or concerns, you will not hesitate to let us know. We are more than willing to provide additional clarifications and supporting materials to ensure the paper meets the highest standard. Your insights are crucial for refining our research and ensuring its relevance and impact.
> >
> > Additionally, we kindly hope that these updates and clarifications will encourage you to reconsider your evaluation, as they directly address your constructive feedback. Should you have any additional queries or reservations, please feel free to contact us at any time. We are fully committed to addressing all concerns to your satisfaction.
> >
> > Thank you once again for your invaluable time and support.
> >
> > Best regards,
> >
> > Authors

---

> > > ### Comment · Reviewer_mjpZ · 2024-11-30
> > >
> > > I appreciate the effort you have put into revising the paper. As all of my concerns have been resolved in the revised version, I have increased my score to 8.

---

> > > > ### Author Response · Authors · 2024-11-30
> > > >
> > > > Thank you for your thoughtful feedback and for taking the time to review our revised manuscript. We greatly appreciate your recognition of our efforts to address your concerns, and we are delighted that the revisions have met your expectations. Your support and encouragement mean a lot to us as we continue to refine our work. Thank you once again for your valuable input!
> > > >
> > > > Authors

---

### Author Response · Authors · 2024-11-21
**General Rebuttal / Revision Response**

Dear Reviewers,

We thank the reviewers for the insightful questions and reviews. Your time and effort dedicated to improving our work are truly appreciated.

We have done all the experiments suggested and answered all the questions. All modifications are marked in red color.

Major revisions include:

**1. New ablation study different action influence on CoA performance in Table 2**.  `reviewer wYMh` `reviewer mjpZ`

- Conducted an ablation study different action influence on CoA performance
- Results:   each module is important to the method performance and each module shows better performance than other baselines.

**2. New ablation study on Latency and LLM usage per question in Table 3.** `reviewer wYMh` `reviewer mjpZ`

- Conducted an ablation study to evaluate the CoA Latency and input and output tokens with baselines
- Results: our method show a less latency and less LLM usage compared with most of baselines.

**3. New experiment on performance without tabular data in Table 6.**  `reviewer mjpZ`

- Conducted an ablation study to performance without tabular data
- Results: From the updated results, we observe a significant decline in the coverage and overall quality metrics without the inclusion of real-time market prices, highlighting the lack of truly useful references. However, the non-redundancy and readability metrics show negligible differences.

**4. New expert recruitment process and questionaire in Appendix F**: additional expert evaluator for human evaluation `reviewer mjpZ`

**5. Revised abstract**:  make the description of the methodology more clear.   `reviewer mjpZ`

**6. Revised Caption in Figure 1**: `reviewer mjpZ`

**7 . Revised in Sec 1**: make the description of the methodology more clear and reclassify the contribution of our work `reviewer mjpZ`

**8. Revised in Sec 2**: make the description of the methodology more clear   `reviewer mjpZ`

**9. Revised metrics in Sec 3.1**: update the definition of ROUGE and ROUGE-L metric used in our paper `reviewer mjpZ`

**10. Revised Implmentation in Sec 3.1**:  refined the claim regarding the "plug-and-play" feature to provide a more accurate description  `reviewer mjpZ`

**11. Revised Caption in Table 3**: add the new experiment of the latency and input &ouput tokens of CoA  `reviewer wYMh`  `reviewer mjpZ`

**12. Revised Caption in Table 6**: add the new experiment of the performance without tabular action `reviewer mjpZ`


We hope these revisions address the reviewers’ concerns and improve the overall quality of our paper.

Thank you again for your review!

Best regards,

Authors

---

Below, we also summarize the key points in our responses:

### Key Points in Our Responses

**Reviewer mjpZ**

* We addressed the concern regarding the accuracy of terminology and methodology descriptions by refining ambiguous terms like "multimodal" and "actions workflow." For example, we replaced "multimodal data" with "heterogeneous data" and clarified the meaning of "knowledge boundary" in response to Question 3. These revisions aim to improve the paper's precision and align terminology with its intended meaning, ensuring clarity for readers.
* We clarified the decomposition strategies critical to our framework by comparing CoA with established baselines in Table 2. Additionally, Appendix D provides a case study to illustrate the distinctions between CoA and CoT. Our analysis of 50 Social QA dataset questions showed CoA achieved 96% correctness and 98% relevance, outperforming CoT. These updates demonstrate the framework's effectiveness and strengthen the contributions of the paper.
* We refined the description of the "plug-and-play" feature by specifying that, while pre-designed actions can be easily added or removed, integrating new actions requires careful design to maintain compatibility with input-output formats. This clarification, included in the introduction, provides a more accurate representation of the feature’s complexity while retaining its flexibility.
* We conducted an ablation study to evaluate the specific contribution of tabular data to our framework. The results, presented in Table 6, show that excluding tabular data reduces coverage and quality metrics, emphasizing its importance in retrieving real-time market data. However, non-redundancy and readability metrics were minimally affected. These findings clarify tabular data's role and address the concern about its impact.

---

> ### Author Response · Authors · 2024-11-21
> **General Rebuttal/Revision Response - Continued 1**
>
> * We expanded the details on the expert evaluation process to enhance transparency and credibility. The experts were selected based on a rigorous survey of Web3 practitioners, and the process is now described in Appendix Section F. This includes criteria, qualifications, and the questions used during evaluation. These additions ensure readers understand the reliability of the evaluation methodology.
> * We clarified the definition of "imputation" in Table 2, referring to filling unanswered sub-questions with retrieved information. Additionally, we renamed CoA-MRFS as CoA(MRFS in verification) to clearly indicate its scope in verification. We also clarify the difference between ROUGE in Table 2 and ROUGE-L used in ASQA long-form QA dataset. These adjustments address confusion around terminology and results in Table 2.
> * We addressed the reviewer’s suggestion to include separate statistics for input and output tokens and even the latency in Table 3. The revised table now provides detailed statistics, including average token usage per action and corresponding time costs. These updates provide a comprehensive understanding of cost details, aligning with the reviewer's request.
> * We clarified "knowledge boundary" to refer to the parametric knowledge the model has learned during training. Leveraging this boundary reduces unnecessary retrieval efforts in RAG-based frameworks, optimizing token usage and LLM interactions. This explanation is now included in the introduction for clarity.
> * We ensured comparability between CoA and previous studies by using the same model (gpt-3.5-turbo) across all baselines for answer generation. GPT-4 was only used for evaluation purposes, as clarified in Section 3.1. This ensures a fair and consistent experimental setup.
> * We updated the section titles in 2.2.1 and 2.2.2 to "Data Collection" and "Data Verification" to align with their content. This change simplifies comprehension and better reflects the focus of the corresponding sections.
>
> ---
>
> **Reviewer wYMh**
>
> * We addressed the concern about the potential latency introduced by CoA by adding the average response time and token consumption details for all baselines in Table 3. These updates show that CoA reduces unnecessary retrievals by leveraging parametric knowledge and employing MRFS for verification, while engineering optimizations, like parallel processing, further minimized response time for real-time applications.
> * We conducted an ablation study to evaluate the impact of removing specific actions, such as web search and local knowledge base search, on performance. The results, added to Table 2, demonstrate that web search contributes more significantly to performance improvements, particularly for non-factual and open-ended QA scenarios, providing clarity on the importance of each action type.
> * We compared CoA with and without external "plugs" across different question types, showing that CoA significantly improves performance in complex scenarios like cross-domain transitions and long-form answers. In contrast, the improvements are modest for simpler, commonsense questions, highlighting CoA's adaptability and effectiveness in handling diverse challenges.
> * We clarified the key differences between "thoughts" in CoT and "actions" in CoA, emphasizing that CoT relies solely on internal reasoning while CoA integrates external tools for complex tasks. This distinction enables CoA to address real-world scenarios with richer, more dynamic reasoning but introduces additional latency and complexity, balanced against its superior capability for nuanced problems.
> * We demonstrated that CoA outperforms CoT even without external actions, as shown in Table 2, by producing more contextualized and comprehensive responses. A detailed case study in Appendix Section D further illustrates CoA’s ability to address nuanced questions effectively, even within the limits of an LLM’s parametric knowledge.
> * We addressed the concern about latency by showing that CoA reduces overall latency, LLM calls, and token costs compared to RAG-based baselines, as reported in Table 3. These optimizations, combined with MRFS verification and knowledge boundary detection, ensure CoA remains efficient and practical for real-time applications while addressing complex information needs.
>
> **Reviewer VgSf**
>
> * We addressed the concern regarding the potential extension of CoA to diverse and unstructured data modalities by outlining these possibilities in response to Question 1. This clarification highlights CoA's adaptability and its feasibility for tasks involving broader data types.
> * We addressed the scalability and efficiency challenges of integrating complex or real-time data sources in response to Question 3. This includes strategies for handling rapidly evolving information to ensure CoA remains efficient and effective in dynamic scenarios.

---

> > ### Author Response · Authors · 2024-11-21
> > **General Rebuttal/Revision Response - Continued 2**
> >
> > * We addressed the concern about CoA's ability to handle tasks involving higher-order reasoning or complex multi-step dependencies in response to Question 4. This discussion explores how CoA is structured to manage intricate reasoning paths, balancing modular design with enhanced reasoning capabilities.
> > * We clarified that CoA currently supports tabular and text data modalities and has been adapted for visual data in our Web3 QA use case. Specifically, we incorporated Vision-Language Models (VLMs) to process cryptocurrency market charts, combining visual insights with local K-line data for enhanced answer reliability. CoA’s decomposition strategy enables seamless integration with VLMs, dynamically generating action chains to address visual or mixed data queries effectively.
> > * We explained that CoA mitigates discrepancies by aggregating content from top-k sources and verifying responses using the Multi-Reference Faith Score (MRFS) to ensure consistency and reliability. Future work will explore advanced methods like voting mechanisms or iterative reasoning to further enhance CoA's ability to reconcile conflicting information while balancing accuracy and latency.
> > * We discussed that CoA is already applied in real-time scenarios, such as Web3 QA, where rapidly changing information is prevalent. Optimization efforts like parallelization and fuzzy search enhance retrieval efficiency, as shown in Table 3, where CoA demonstrates superior latency and cost performance compared to RAG-based frameworks. Future work includes building a real-time benchmark to evaluate CoA's effectiveness in dynamic environments.
> > * We highlighted that CoA can handle intricate reasoning paths through an iterative generation mechanism that refines action chains dynamically. While currently limited to single-generation for efficiency, we included an iterative approach in the appendix as a future direction to enhance CoA's capability for more complex applications without incurring excessive latency.

---

> > > ### Author Response · Authors · 2024-11-27
> > > **Follow-Up on Rebuttal Phase Feedback**
> > >
> > > Dear Reviewers,
> > >
> > > Thank you once again for your valuable feedback. We have carefully addressed your comments and made substantial revisions to improve the manuscript.
> > >
> > > As the discussion phase nears its conclusion, please don’t hesitate to let us know if you have any further questions or concerns. Wishing you a wonderful Thanksgiving, if you celebrate it, and thank you for your time and consideration.
> > >
> > > Best regards,
> > > Authors

---

### Meta-Review · Area_Chair_QTw5 · 2024-12-22

**Metareview:**

This paper introduces the Chain-of-Action (CoA) framework, a new approach to improving multimodal and retrieval-augmented QA for LLMs. The framework tackles two core challenges in QA—unfaithful reasoning and information hallucination—by decomposing complex questions into sequential reasoning steps termed "actions." These actions include web querying, knowledge encoding, and data analysis, enabling systematic integration of heterogeneous data sources. Additionally, the authors propose the multi-reference faith score (MRFS), which cross-validates model outputs against multiple sources to improve reliability. Empirical results demonstrate CoA's effectiveness, showing improved performance across public QA benchmarks. The framework claims advantages in cost-efficiency (via reduced API calls) and flexibility, allowing integration of diverse data types.

Strenghts:
- CoA’s structured decomposition into sub-questions/actions represents a significant advancement in tackling complex QA tasks. Its modular design provides adaptability for various data types and retrieval methods.
- The proposed framework consistently outperforms baseline methods (e.g., CoT, DSP, SearchChain) across multiple QA benchmarks, demonstrating its robustness and versatility.
- The introduction of MRFS as a reliability metric enhances the trustworthiness of responses, mitigating common issues like hallucination and conflicting information.
- CoA demonstrates efficiency in reducing token usage and API calls. This provides clear advantages for scalability in cost-sensitive deployments.

Weaknesses:
The initial weaknesses of the paper raised by reviewers include the following:
- Terminology used in the paper, such as "multimodal," "plug-and-play," and "chain-of-action," are ambiguously defined or potentially misapplied (e.g., text and tabular data being labeled as "multimodal").
- The paper does not include sufficient ablation studies to isolate the contributions of individual components, such as the impact of MRFS or the use of tabular data in performance gains.
- While CoA reduces token usage, its modularity and additional retrieval steps may introduce significant latency, especially with complex or real-time data sources. The study lacks an analysis of response times and scalability for dynamic scenarios.
- The framework's applicability to non-textual or highly unstructured data, such as visual or live-stream data, remains unexplored, which limits its generalizability.

During the rebuttal stage, the authors revised the manuscript and addressed the majority of these weaknesses.

Overall, this paper makes a meaningful contribution in improving multimodal and retrieval-augmented QA for LLMs. The paper received strong ratings with at least two reviewers excited about the paper, and no reviewer opposing acceptance. After evaluating the discussions and the revision, to me it appears that the authors have adequately addressed the major concerns.

**Additional Comments On Reviewer Discussion:**

Reviewer mjpZ acknowledged that the author response effectively addressed all their questions and concerns, which led to an increase in their evaluation scores. Although reviewer VgSf did not provide feedback on the author's responses, the authors have revised the manuscript to address each of the reviewer VgSf' questions and concerns.

---

### Decision · Program_Chairs · 2025-01-22

Accept (Poster)